

# The impact of GPS and high-resolution radiosonde nudging on the simulation of heavy precipitation during HyMeX IOP6

Alberto Caldas-Alvarez[1], Samiro Khodayar[1,2] and Peter Knippertz[1]

[1]Institute of Meteorology and Climate Research (IMK-TRO), Karlsruhe Institute of Technology, Karlsruhe, P.O. Box 3640, 76021, Germany
[2]Mediterranean Centre for Environmental Studies (CEAM), Valencia, 46980, Spain

*Correspondence to* Alberto Caldas-Alvarez (alberto.caldas-alvarez@kit.edu)

## Abstract

Heavy precipitation is one of the most devastating weather extremes in the western Mediterranean region. Our capacity to prevent negative impacts from such extreme events requires advancements in numerical weather prediction, data assimilation and new observation techniques. In this paper we investigate the impact of two state-of-the-art data sets with very high resolution, Global Positioning System-Zenith Total Delays (GPS-ZTD) with a 10 min temporal resolution and radiosondes with ~700 levels, on the representation of convective precipitation in nudging experiments. Specifically, we investigate whether the high temporal resolution, quality, and coverage of GPS-ZTDs can outweigh their lack of vertical information or if radiosonde profiles are more valuable despite their scarce coverage and low temporal resolution (24h to 6h). The study focuses on the Intensive Observation Period 6 (IOP6) of the Hydrological Cycle in the Mediterranean eXperiment (HyMeX; 24 September 2012). This event is selected due to its severity (100 mm/12h), the availability of observations for nudging and validation, and the large observation impact found in preliminary sensitivity experiments. We systematically compare simulations performed with the COnsortium for Small scale MOdelling (COSMO) model assimilating GPS, high- and low vertical resolution radiosoundings in model resolutions of 7 km, 2.8 km and 500m. The results show that the additional GPS and radiosonde observations cannot compensate errors in the model dynamics and physics. In this regard the reference COSMO runs have an atmospheric moisture wet bias prior to precipitation onset but a negative bias in rainfall, indicative of deficiencies in the numerics and physics, unable to convert the moisture excess into sufficient precipitation. Nudging GPS and high-resolution soundings corrects atmospheric humidity, but even further reduces total precipitation. This case study also demonstrates the potential impact of individual observations in highly unstable environments. We show that assimilating a low-resolution sounding from Nimes (southern France) while precipitation is taking place induces a 40 % increase in precipitation during the subsequent three hours. This precipitation increase is brought about by the moistening of the 700 hPa level (7.5 gkg$^{-1}$) upstream of the main precipitating systems, reducing the entrainment of dry air above the boundary layer. The moist layer was missed by GPS observations and high-resolution soundings alike, pointing to the importance of profile information and timing. However, assimilating GPS was beneficial for simulating the temporal evolution of precipitation. Finally, regarding the scale dependency, no resolution is particularly sensitive to a specific observation type, however the 2.8





km run has overall better scores, possibly as this is the optimally tuned operational version of COSMO. In follow-up experiments the Icosahedral Nonhydrostatic Model (ICON) will be investigated for this case study to assert whether its numerical and physics updates, compared to its predecessor COSMO, are able to improve the quality of the simulations.

## 1 Introduction

In the Western Mediterranean Heavy Precipitation Events (HPEs) cause fatalities and large economic losses every year (Petrucci et al., 2018). Many of these events occurs during autumn, since this is the time when large-scale systems (northerly troughs, extratropical cyclones) coincide with large mesoscale moisture transports, advected with the southerly flow (Toretti et al., 2010; Pinto et al., 2013; Dayan et al., 2015; Gilabert and Llasat, 2017). This creates the propitious humidity and instability conditions for convective systems that get triggered due to orography, wind convergence or thermodynamic processes (Ricard et al., 2012; Khodayar et al., 2016). The representation of such events is still a challenge for current Numerical Weather Prediction (NWP) models. The relatively short temporal and spatial scales of convective phenomena, from minutes to less than a day and from $10^0$ to $10^2$ km (Markowski and Richardson, 2010), make it challenging to forecast accurately where and when an HPE will develop. One of the identified sources of error is the misrepresentation of the spatial distribution of atmospheric moisture. Current models have shown a large sensitivity of convective precipitation to small differences in moisture distribution (Lintner et al., 2017, Virman et al., 2018). Hence, it is hoped that using sub-kilometre resolution and sub-hourly frequencies in atmospheric models and Data Assimilation (DA) systems can lead to substantial improvements for HPE prediction.

DA is a key ingredient to the initial value problem of NWP (Bauer et al., 2015), as the frequent assimilation of high-quality observations helps adjust the NWP model towards the true atmospheric state. Recent advancements in observation systems and high-performance computing have brought progress for DA of new observations (Carlin et al, 2017; Kwon et al., 2018; Borderies et al., 2019; Federico et al., 2017, Mazzarella et al., 2017). GPS measurements of ZTDs are an especially interesting observation type, since they can sample the Integrated Water Vapour (IWV) amount at minute temporal-resolution (Bock et al., 2016). Past publications have shown improvements of GPS data for screen-level temperature and humidity (Mile et al., 2019; Mascitelli et al., 2019), lower to middle tropospheric moisture (Singh et al., 2019; Bastin et al., 2019; Caldas-Alvarez and Khodayar, 2020) and 24-hour precipitation (Hdidou et al., 2020; Fourrié et al., 2020). The advantages of this product are its high temporal resolution, that it is all-weather (as opposite to satellite radiances or ground radars), has large accuracy (Bock et al., 2016, Bock et al., 2019; Jones et al., 2019) and has a dense coverage over European countries. However, being an integrated quantity, GPS measurements bear no information of the vertical distribution of humidity (Guerova et al., 2016). This deficit can be decisive in some situations since sufficient moisture even in shallow atmospheric layers can make the difference between convective triggering or suppression. For example, Davolio et al. (2017) find a good impact on precipitation forecasts when restricting moisture corrections to the boundary layer. Federico et al. (2019) show that assimilating derived humidity profiles from radar reflectivity and lighting data induces a moist bias possibly due to the inability of the system to





correctly redistribute the humidity into shallower vertical layers. This is one of the reasons why, to date, radiosondes have
remained the backbone of DA systems (Kwon et al., 2018). Radiosondes can supplement the lack of vertical information of
GPS observations, however at the expense of a coarser temporal resolution (in best cases, every 6 h) and lower spatial coverage
(~ 30 stations in western Europe). To bring the best value from radiosonde assimilation, targeted observations can help sample
atmospheric conditions at the right place and time, e.g. in regions upstream of areas prone to heavy precipitation (Campins et
al., 2013). In this regard, an open question remains as to how high the vertical resolution of the soundings needs to be to obtain
optimal improvement.

Past modelling and observational studies demonstrated that a good account of the spatial distribution of atmospheric moisture
is crucial for the representation of convective intensities (Keil et al., 2008, Lintner et al., 2011, Honda et al., 2015, Schumacher
et al., 2015, Schlemmer et al., 2015, Lintner et al., 2017, Virman et al., 2018). Consensus exists that a moist Planetary Boundary
Layer (PBL) is needed for convective triggering and maintenance (Lee et al., 2018). However, the dependency on Lower Free
Tropospheric (LFT) moisture is less established. Dry mid-levels can lead to a faster organisation of mesoscale clusters through
stronger cold pools (Zuidema et al., 2017) but also reduce the strength of connective updrafts through entrainment of drier air.
Several authors have highlighted the important role of mid-level moisture in aiding the transition from shallow to deep moist
convection (Lintner et al., 2011; Neelin et al., 2009; Bernstein et al., 2016; Zhuang et al., 2018; Khodayar et al., 2018), as a
sufficiently moist LFT prevents excessive entrainment (Honda et al., 2015), helps convection penetrate possible intrusions and
maintains the buoyancy of the rising parcels.

In addition to DA, convection-permitting model resolution has brought important advancements in the simulation of heavy
precipitation (Chan et al 2012; Prein et al., 2015; Coppola et al., 2018; Meredith et al., 2020). However, the question regarding
how fine model resolutions should be, beyond the kilometre scale, in the so-called grey-zone (Barthlott and Hoose; 2015) is
still open. Several papers have found improvements from using model resolutions of 1.5 km or higher (Kendon et al., 2012;
Martinet et al., 2017; Bonekamp et al., 2018 and Lovat et al., 2019), whereas others found no significant gain (Chan et al.,
2012; Panosetti et al., 2016; Lee et al., 2019). A possible reason is the fact that the appropriate settings for running current
models at such high resolutions are not yet ready. In this context it is interesting to assess the sensitivity of the impact of new
observations such as GPS and high-resolution soundings to model resolution.

Assessing the capabilities of current NWP systems for heavy precipitation is one of the aims of HyMeX, an international
project aiming at a better understanding of the hydrologcal cycle in the Mediterranean (Ducrocq et al., 2014). The Special
Observation Period 1 (SOP1) between 01 September and 05 November 2012 provides an unprecedented collection of data that
is used in this study for assimilation, validation, and process-understanding. The event we focus on occurred on 24 September
2012, during Intensive Observation Period 6 (IOP6), and brought precipitation amounts of over 100mm in 12h to southern
France, the Alps, the Gulf of Genoa, and northeastern Italy (Hally et al., 2014; Ribaud et al., 2016). This HPE showed a
negative impact of GPS DA in preliminary assimilation tests, related to an overall reduction of atmospheric moisture and
precipitation amount (between -40% and -10% depending on model resolution). Given this unexpected result, we will
investigate here in more detail which characteristics of the GPS DA were detrimental for the representation of precipitation.





To do so, we will systematically assess the impact of nudging GPS, operational soundings, and high-resolution soundings using COSMO simulations with – for this case unprecedently fine – grid-spacings of 7 km, 2.8 km and 500m. The employed

nudging scheme (Schraff and Hess, 2012) is well suited for such studies (Federico et al., 2019; Bastin et al., 2019) and compares well against other DA schemes (Schraff et al., 2016). The employed methods will be outlined in Section 2. Section 3 discusses the meteorological situation during IOP6 and the model runs used as reference. Section 4 presents the results of the data impact studies. Conclusions are given in Section 5.

## 2. Data and Methods

**2.1 Observations**

### 2.1.1 GPS-derived Zenith Total Delays (ZTD) and Integrated Water Vapour (IWV)

The GPS data set used for the nudging was specifically produced for the HyMeX project, merging measurements from 25 European national and regional networks commonly post-processed for the first time to cover the period September 2012 to March 2013 (Bock et al., 2016). GPS measurements provide information of the total delay endured by the microwave signals

emitted by GPS satellites in the Zenith direction (Zenith Total Delay; ZTD). These are expressed in units of mm, accounting for the excess length of the optic path introduced by the refractivity of the Earth's atmosphere (Businger et al., 1996). The contribution to the delay due to the interaction with water vapour molecules is called the "wet" delay and can be obtained from the ZTD. This is the assimilated variable in the nudging experiments, which is proportional to the IWV. The data set used in this paper has a temporal resolution of 10 minutes, an outstanding spatial coverage over all southwestern European countries

(see Fig.1b) and was produced using the GIPSY/OASIS II v6.2 software (Bock et al., 2016). It enjoys a very high quality due to its data screening procedure, including range and outlier checks for mean ZTD and its standard deviation, as well as ambiguity and daily number checks. Compared against the product from the Network of European Meteorological Services (EUMETNET) Global Navigation Satellite System (GNSS) Water Vapour Programme (E-GVAP), the HyMeX data set shows no significant biases (Bock et al., 2016).

HyMeX, also provides an IWV data set with 1h resolution, derived from the ZTD estimations (Fig. 1b). We employ this IWV data set for comparison against our simulations. The mean temperature and surface pressure values at the GPS station locations, which are needed for the IWV derivation, were obtained from a product provided by the Technical University of Vienna and AROME western Mediterranean operational analysis, respectively. A validation of the IWV product against operational radiosondes showed a good performance, with biases of less than 1.5 mm for the whole HyMeX period (Bock et al., 2016).

### 2.1.2 The operational and the HyMeX high-resolution soundings

Operational atmospheric sounding data are provided by Météo-France and the HyMeX database teams through the HyMeX-MISTRALS web repository (https://mistrals.sedoo.fr/HyMeX/). The data set consists of atmospheric soundings during the



period 1995–2017, covering the western Mediterranean countries (blue triangles in Fig. 1b), operated by national and regional European atmospheric weather institutions and distributed through the Global Telecommunication System (GTS). The

soundings have 30 vertical levels on average and have been validated against GPS measurements with good agreement. Deviations of only ± 3 % in IWV were found by Bock et al. (2016) for the soundings.

In addition, we employed the unique high-resolution soundings of the HyMeX SOP1 in the nudging experiments. These were conducted at locations upstream of areas prone to heavy precipitation (red squares in Fig. 1b). They have a much finer vertical resolution with over 700 levels up to 300 hPa. We employed soundings from twelve stations over France (continental and

Corsica) and Spain. Modem-M10 sondes were launched at Ajaccio (Corsica), Nimes, and Barcelona, Graw sondes, operated by the Karlsruhe Institute of Technology, at Corté and Inra (Corsica), and Vaisala sondes in southern France and Spain.

### 2.1.3 Meteosat Second Generation (MSG) Brightness Temperature

Brightness temperature is an estimation of the radiation emitted by a surface, converted to temperature through Planck's law, assuming a black body. It provides a clue of the height of cloud tops and is especially useful for deep penetrating convective

clouds. In this paper we use the All-Sky radiances product, obtained by the Spinning Enhanced Visible and InfraRed Imager (SEVIRI) instrument on-board the Meteosat Second Generation satellite constellation. In particular, the InfraRed (IR) channel IR10.8 is used for detection of organized convective systems. The data are accessible upon registration at https://www.eumetsat.int/website/home/Data/DataDelivery/index.html.

### 2.1.4 Rain Gauges (RG) and Multi-Source Weighted-Ensemble Precipitation (MSWEP)

The RG data set used for validation in this paper is available for accumulation periods of 1h, 6h or 24h, has a dense coverage of Spain, France, Italy, and Croatia, and on average over 4000 stations active per sampled hour. The data set is made available by Météo-France by means of the MISTRALS/HyMeX repositories. The version used for this study is V4, which includes high-resolution measurements from Italy and Croatia as compared to older versions. Several quality checks are included in this version, such as consistency validations among the different accumulation periods, removal of duplicates and dismissal of

blacklisted stations.

The MSWEP product is used for validation of our model results. We use version V2.1 with a temporal resolution of 3h and a spatial resolution of 0.1° during 1979–2015. We examine the period 22–25 September 2012. A full description of the data set can be found in Beck et al. (2017). MSWEP is a gridded precipitation dataset merging satellite, re-analysis and gauge-based estimates, utilizing, among others, the Climate Prediction Center Morphing Technique (CMORPH), Precipitation Estimation

From Remotely Sensed Information Using Artificial Neural Networks (PERSIANN), Tropical Rainfall Measuring Mission (TRMM), ERA-interim (reanalysis) and Climate Prediction Center (CPC) and Global Precipitation Climatology Center (GPCC; gauge) observations. The MSWEP product shows a good correlation with the independent FLUXNET gauge network with median values of 0.65 for the Pearson correlation coefficient. RMSE median values were of 4.5 mm d$^{-1}$, showing better





results than TRMM TMPA 3B42 or WFDEI-CRU (Beck et al. 2017). We selected this precipitation data set for model

validation given it profits from the combined value of satellite precipitation products as well as RG.

## 2.2 The COSMO model

COSMO uses the non-hydrostatic, thermo-hydrodynamical equations in a limited area approach (Schättler et al., 2012), considering the wind components, temperature, pressure perturbation, the cloud water content, and the specific humidity as prognostic variables. Optionally, also cloud ice, snow and graupel can be considered (Schättler et al., 2012). The model levels

are based on a height coordinate that follows the terrain. The rotated grid is an Arakawa C type with Lorenz vertical grid staggering. The dynamical solver is a second order leapfrog time-split scheme after Skamarock and Klemp (2002). COSMO includes physical parameterizations for the processes that are not explicitly represented. The grid-scale clouds and precipitation parameterization uses a bulk scheme continuity model including water vapour, cloud water, cloud ice, rain, snow and graupel as water species. Convection is parameterized using the Tiedtke scheme (1989), a bulk-mass-flux formulation dependent on

mass, heat, moisture, and momentum fluxes, including a cloud model, simulating processes such as condensation/deposition, evaporation within the updraft and evaporation below cloud base. The radiation scheme follows the Ritter and Geleyn description (1992) and is applied with a lower temporal frequency and lower resolution than that of the rest of the model to reduce computational costs. The soil model is the Terra Multi-Layer (ML) model that is based upon the two-layer scheme by Jacobsen and Heise (1982). Finally, the surface data uses the GLOBE dataset (Hastings et al., 1998) with a 1 km resolution

adequately interpolated (extrapolated) to the scale of the different resolutions used (7 km, 2.8 km, and 500m).

One of the main assets of COSMO is its flexibility to be used with different horizontal resolutions, each of which requires specific configuration settings. For finer spatio-temporal scales, more processes are explicitly resolved at the expense of higher computational costs. In this work we employ horizontal grid spacings of 7 km, 2.8 km, and 500m. The most relevant differences between the 7 km and the 2.8 km set-ups are (a) the increase of levels from 40 to 50, (b) the reduction of time step from 60s

to 20s, and (c) the use of only a shallow parametrization scheme in 2.8 km. The formulation of the latter is analogous to the deep scheme, except for the reduced vertical extent of clouds (limited to $\Delta p = 250\ hPa$; Baldauf et al., 2011) and the neglection of dynamic entrainment (Doms et al., 2011). This scheme is inactive in sub-kilometre simulations, i.e. in our 500m simulation. Other changes are (a) a further increase of vertical levels to 80, (b) a time step of 2s, and (c) the use of a 3D Turbulent Kinetic Energy (TKE) diagnostic closure for the turbulence parametrization. In the 7 km and 2.8 km configurations,

the closure of the turbulence parametrization scheme is 1D in that it neglects all horizontal fluxes in the so-called boundary-layer approximation (Doms et al., 2011). In the 3D TKE closure case, the vertical shear production term can come from local sources as well as from advection, and the pressure correlation term is explicitly calculated, which is especially useful over complex terrain (Goger et al., 2018).



### 2.2.1 The COSMO Nudging scheme

The DA method used in this work is the Nudging scheme (Schraff and Hess, 2012). Nudging is an empirical DA method consisting of relaxing the model's prognostic variables towards the observations. This is done by adding a term for a given prognostic variable ($\varphi$) that depends on the difference between the observation (k) and the model ($\varphi_k^{obs} - \varphi^{mod}(x_k, t)$), the temporal, spatial and quality weighting factors ($W_k(x, t) = \left(w_k / \sum_j w_j\right) \cdot w_k$), depending in turn on a relative weight for each observation type ($w_j$) and a coefficient with units of frequency ($G_\varphi$), see Eq. 1. The nudging is performed at each model time

step when observations are available.

$$\frac{\partial}{\partial t} \varphi(x, t) = F(\varphi^{mod}, x, t) + G_\varphi \cdot \sum_k W_k(x, t) \cdot \left[\varphi_k^{obs} - \varphi^{mod}(x_k, t)\right] \qquad [1]$$

In this work, we nudge atmospheric specific humidity (GPS and radiosondes), temperature (radiosondes) and wind (radiosondes). These are the quantities assimilated operationally at forecasting centres from GPS and radiosonde measurements (Kwon et al., 2018). The nudging scheme is especially suited for these experiments, since it corrects the atmospheric fields

during run time, with the same frequency as the sampling of observations. Additionally, it has shown good results in analysing humidity fields, especially at upper levels (Schraff et al., 2016; Bastin et al., 2019) and is computationally less expensive than other schemes (Variationals or Hybrids) given its simplicity (Guerova et al., 2016).

### Nudging of GPS and radiosondes

The COSMO nudging scheme only allows the assimilation of prognostic variables. In the case of the radiosondes, COSMO

reads profiles of temperature, wind and humidity assigning all observations to a grid point in model space. Given that the grid points cannot correctly represent wavelengths of $2\Delta x$ or less, the assignment is performed with no interpolation in the horizontal direction (Schraff and Hess, 2012). The impact of the analysis increments on the neighbouring grid points is controlled through lateral ($w_{xy}$), vertical ($w_z$) and temporal weights ($w_t$) through the equation $w_k = w_{xy} \cdot w_z \cdot w_t \cdot \varepsilon_k$, where $\varepsilon_k$ accounts for the quality and representativeness of the observation. At the exact time-space location of the observation $w_{xy}$,

$w_z$ and $w_t$ are set to 1.

The temporal spreading is controlled by the nudging coefficient, which is currently set to $6.10^{-4}$ s$^{-1}$, this corresponds to an e-folding decay of half an hour. For radiosondes the assimilation time window is set between -3 and +1 hours. The vertical spreading weight follows a Gaussian in height differences, accounting for the hydrostatic relation and the ideal gas law (Eq. 2).

$$w_z = exp\left\{\frac{-[g/R\,T_v|\cdot \Delta z]}{ln p_c}\right\}^2 \qquad [2]$$

Horizontally, the spreading is performed using a second-order autoregressive function of the distance between the observation location and the target point ($\Delta r$) divided by correlation scale (s), see Eq. (3). The impact of the assimilated observation decreases with the distance to the station location.

$$w_{xy} = (1 + \Delta r/s) \cdot e^{-\Delta r/s} \qquad [3]$$





To assimilate GPS data, COSMO converts the ZTD information into IWV (see Section 2.1.1) utilizing simulated $p_s$ and $T_m$ at run time from the assigned grid point. Given IWV is not a prognostic variable, a specific humidity profile needs to be constructed ($q_v^{mod}$). This is done by means of an iterative process that scales the IWV simulated at that location and time ($IWV^{mod}$) with that of the observation ($IWV^{obs}$). The profile is constructed at the different model levels according to

$$q_{v_{i+1}}^{mod} = q_{v_i}^{mod} \cdot \frac{IWV_{i+1}^{obs}}{IWV_i^{mod}} \quad . \tag{4}$$

The process continues until the IWV error is lower than 0.1 % or after 20 iterations (Schraff and Hess, 2012). In the remainder of the process the constructed profile is treated in the same way as the one derived from radiosondes with the exceptions that (a) no vertical weights are needed, since the profile is constructed over model levels directly and (b) temporally, GPS data are interpolated linearly given their minute temporal resolution. Both for radiosonde and GPS observations, the nudging scheme sequentially carries out quality checks for new input observations. These checks consist of dismissal of observations with large
biases; bias corrections, e.g. humidity biases in Vaisala soundings, and gross error checks to truncate the range of the observations within realistic limits.

### 2.3 Experimental set-up

We run 3-day simulations between 22 September 2012 0000 UTC and 25 September 2012 0000 UTC. We simulate this period with COSMO in three different configurations, using horizontal resolutions of 7 km, 2.8 km, and 500m in a one-way nesting
strategy (Fig. 1a). Integrated Forecasting System (IFS) simulations from the European Centre for Medium-Range Weather Forecasts (ECMWF) with a resolution of 25 km force the 7 km runs, which in turn force the 2.8 km and finally these force the 500m simulations. The following rationale guides the nudging experiments. The study period is run in all three resolutions as pure forecast runs (named CTRL-7, CTRL-2.8, CTRL-500), which are used as references to compare against simulations nudging GPS, operational RADiosondes (RAD) and High-Resolution radiosondes (HR) and all possible combinations between
them (see Table 1), in all three resolutions. The nudging is performed continuously processing new observations as soon as they are available for the time step under integration. This implies that the average frequency for nudging of GPS is 10 minutes and between 6 and 12h for radiosondes. We use ca. 1000 GPS stations, 32 RAD sounding stations and 12 HR stations (Fig. 1b). All simulations of the same resolution are forced with the same boundary conditions. For instance, all 500m simulations are forced by CTRL-2.8. This is done to ensure that the different impacts observed in the simulations are due to the use of
different observations and not from different forcing data. A total of 21 simulations were performed (see Table 1). The study focuses on two investigation areas, the Cévennes-Alpine area in southern France (FR) and the north western Mediterranean basin (RhoAlps), see Fig.1b. The extension of FR has been selected for study of local instability, moisture and wind conditions influencing convective activity over the area. RhoAlps covers the extension of the four main heavy precipitation foci (see Fig. 2b).



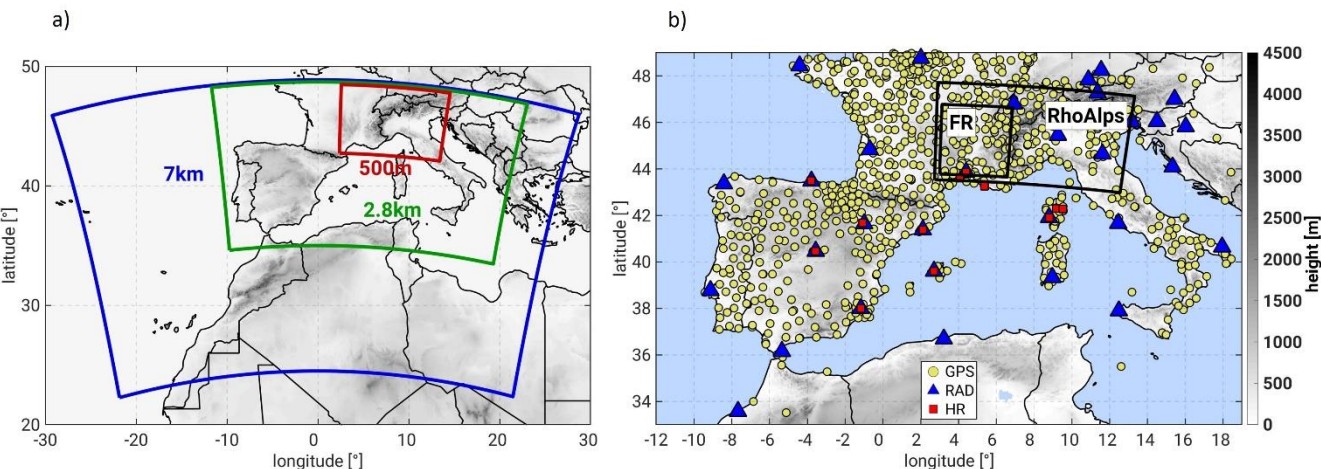


**Figure 1: (a) Nested simulation domains for the different resolutions. (b) Spatial distribution of nudged measurements, GPS, operational RADiosoundings (RAD) and High-Resolution soundings (HR), and investigation domains FR and RhoAlps (boxes).**

**Table 1. Summary of simulations nudging radiosondes and GPS observations. In total 18 simulations with nudging plus three control runs were performed. The simulations with combined nudging of observations maintain the same assimilation frequency and**
**number of levels for the different observations. The resolution of the GLOBE orography data set is 1 km. The TERRA-ML model is used for the soil atmosphere interactions parametrizations.**

| Resolutions | | | | | | Observations | | | Constant Parameters |
|---|---|---|---|---|---|---|---|---|---|
| | Forcing | Lev. | Convec. | Turb. | | | Freq. | Levels | |
| 7 km | IFS | 40 | Tiedtke Deep | 1D TKE | 3x7= | RAD (Oper. Rads.) | ~ 6 h | ~20 | GLOBE-1km (orography) |
| | | | | | **21 sims**. | HR (High-res. Rads.) | ~6 h | ~4000 | |
| 2.8 km | CTRL-7 | 50 | Tiedtke Shallow | 1D TKE | ⟷ | GPS | 10 min | Integr. | |
| | | | | | | GPS-RAD | Combined instruments | | TERRA-ML (soil) |
| 500m | CTRL-2.8 | 80 | - | 3D TKE | | RAD-HR | | | |
| | | | | | | GPS-RAD-HR | | | |
| | | | | | | CTRL (No obs.) | | | |

## 2.4 Verification metrics

We validate the timing of area-mean precipitation using anomaly correlations and the structure of precipitation obtaining the Fractions Skill Score (FSS) compared to gridded data from MSWEP. Area and time averaged water vapour, both integrated
and at each pressure levels, are validated by means of the Root Mean Square Error (RMSE) and the Mean Bias (MB).





### 2.4.1 Anomaly correlation

In Section 4.1, we validate the anomaly correlation for precipitation between the different simulations. To this end, we obtain and subtract the mean precipitation during 24 September 2012, spatially averaged from the time series. This is done separately for the observations and the model. Then, for each temporal evolution, we calculate the Pearson's correlation coefficient
between the model and observations (Joliffe and Stephenson, 2011).

$$r_{mod,obs} = \frac{\sum_{i=1}^{24h}(mod_i - \overline{mod})(obs_i - \overline{obs})}{\sqrt{\sum_{i=1}^{24h}(mod_i - \overline{mod})^2}\sqrt{\sum_{i=1}^{24h}(obs_i - \overline{obs})^2}} \qquad [5]$$

The spatial averaging is performed over the investigation area RhoAlps, where only land points are considered due to the lack of data of MSWEP over the sea; all simulations are coarse-grained to the MSWEP resolution.

### 2.4.2 Fractions Skill Score (FSS)

The FSS provides an estimate of the agreement in the fraction of surface affected by precipitation between observations and simulations. After coarse-graining the simulations output to the resolution of the observations (MSWEP, 0.1°), each grid point within the investigation area (both for observations and simulations) is given a binomial value depending on daily precipitation. A value of 1 is assigned to grid points with precipitation over 20 mmd$^{-1}$ and 0 is given to the remainder grid points. We selected this threshold to be able to have defined precipitation structures within the investigation area following Robert and Lean (2008)
and Skok et al. (2016). For each grid point, the fractions of affected surface are calculated for a number of near neighbours as $f = n_{precip}/n_{tot}$. Where $n_{precip}$ are the grid points affected by precipitation (binomial value of 1) and $n_{tot}$ is the total number of neighbours. In our study we used 18 neighbours, the maximum number of grid points in the zonal direction of the RhoAlps investigation area. To select this threshold we follow Robert and Lean (2008) and Skok et al. (2016), which demonstrated that the skill of a forecast, as quantified by the FSS, depends on the number of neighbours considered with the largest skill being
brought by a computing FSS with the largest number of neighbours. FSS is then computed as

$$FSS = 1 - \frac{\frac{1}{N}\sum_{i=1}^{N}(f_{mod} - f_{obs})^2}{\frac{1}{N}(\sum_{i=1}^{N}f_{mod}^2 + \sum_{i=1}^{2}f_{obs}^2)} \qquad [6]$$

The best representation is shown with values closer to one. Given the characteristics of this score, there is a limit FSS value below which a forecast has no skill. This target skill depends on the fraction of grid points affected by precipitation in the observational data set ($f_{obs}$). The target skill is defined as $FSS_{target} = 0.5 + f_{obs}/2$ and is denoted by a dashed line in Fig.5c.

### 2.4.3 Root Mean Square Error (RMSE) and Mean Bias (MB)

The validation of IWV and specific humidity simulated with COSMO is quantified through the RMSE and MB (Eq. 7 and 8) in Section 4.2, where $i$ is the running index for all available observations (N):

$$RMSE = \sqrt{\frac{1}{N}\sum_{i}^{N}(mod_i - obs_i)^2} \qquad [7]$$





$$MB = \frac{1}{N}\sum_{i}^{N}(mod_i - obs_i) \qquad\qquad [8]$$

### 3 The HyMeX IOP6 (24 September 2012)

In the night of 24 September 2012 several Mesoscale Convective Systems (MCSs) were active over southern France, the Alps and the Italian Gulfs of Genoa and Venice (Hally et al., 2014; Ferretti et al., 2014). Over the course of 12h, RG recorded totals as large as 100 mm over Montélimar, the Swiss Alps and at the Austrian-Italian border (Fig 2b). In total four regions can be characterized by heavy precipitation: the Rhone valley (France), Lugano (Switzerland), La Spezia (Italy) and Udine (Italy).
The synoptic situation was dominated by an upper-level trough situated over western Europe and a surface low to the northwest of Ireland during the night of 23 September 2012 (Hally et al., 2014; Taufour et al., 2018). The associated cold front moved over southern France, the Alps and north-eastern Italy in the course of 18 hours, as the surface low moved from Ireland to the Baltic Sea. A squall line developed over southern France at 0000 UTC on 24 September 2012 (Fig. 2a), reaching its mature phase at 0300 UTC and splitting into two smaller MCSs at 0500 UTC (Ribaud et al., 2016). The MCSs moved from north-western to north-eastern Italy after midday (Pichelli et al., 2017; Fig. 2a). The cyclonic circulation swept in air from the Mediterranean over the Gulf of Lions, the Gulf of Genoa and up to Venice through the Adriatic Sea (Hally et al., 2014). The additional low-level moisture supported the unstable conditions needed for convective development and fed the active systems until their decay after 2000 UTC on 24 September 2012.

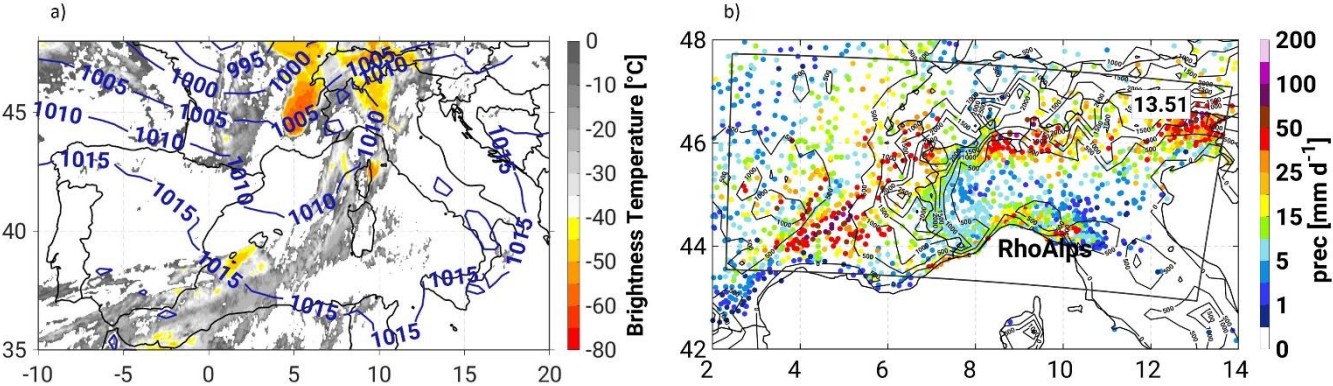

**Figure 2. Synoptic overview of IOP6. (a) Brightness temperature of the MSG-SEVIRI instrument channel 9 (MSG-0 degree, all-sky radiances, https://www.eumetsat.int/) and isolines of mean sea level pressure simulated by COSMO (7 km) on 24 September 2012 at 0600 UTC. (b) 24-hourly accumulated precipitation from the HyMeX RG data set. The dark boxes FR and RhoAlps denote the investigation areas.**

COSMO is able to represent the event, capturing the four main precipitation spots and the main features such as the squall line addressed by Hally et al. (2014). To demonstrate this, Fig. 3 represents the spatial distribution of 24-hourly aggregated rainfall simulated by COSMO (Figs. 3b–d) and estimated by MSWEP (Fig. 3a). Overall, MSWEP represents well the event over the RhoAlps area albeit clear differences in structure due to the coarser resolution of ca. 10 km. MSWEP also overestimates precipitation north of the Rhone valley and over the Alps with totals of up to 50 mm, not measured by RG. Regarding COSMO,

the precipitation intensities stay within the observed range despite a tendency for underestimating the 24-hourly aggregations

(Figs. 2b and 3). Irrespective of resolution, COSMO struggles to represent the observed intensities as large as 100 mm. Differences also occur in the precipitation structure and location with some dependency on the model resolution. CTRL-7 shows the location of the convective line over FR shifted towards the Alps and a too narrow and intense precipitation structure over the Udine maximum. CTRL-2.8 shifts the precipitation maxima over FR northward and splits the Udine maximum into two, one over Udine and the other one over the Gulf of Venice. Finally CTRL-500 represents a narrower convective line over

FR with a better agreement with observations and as CTRL-2.8, a split maximum over north-eastern Italy.

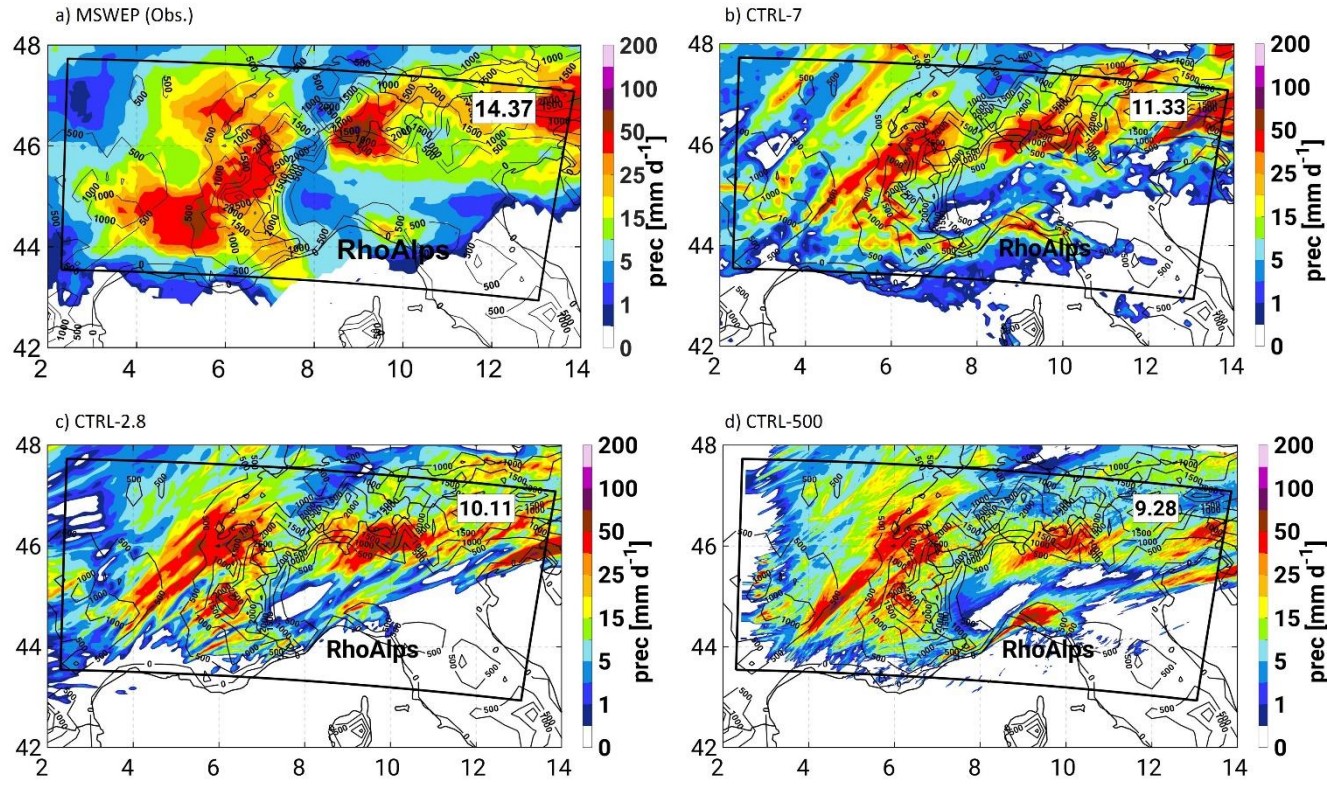

**Figure 3. Spatial distribution of daily precipitation during 24 September 2012 0000 UTC to 25 September 0000, estimated by the MSWEP v2.1, 3-hourly, 0.1° (a) and simulated by CTRL-7 (b), CTRL-2.8 (c) and CTRL-500 (d). The labels within the RhoAlps box**
**shows the values of the spatial averages used in Fig. 5a for validation of the precipitation totals.**

## 4 Impact of GPS, operational RADs and HR radiosonde nudging

### 4.1 Precipitation

The different observation types impact precipitation independently of the model resolution. Figure 4, analogously to Fig. 3, shows the spatial distribution of 24-hourly aggregated rainfall. In this case only the 500m resolution is shown given the





similarities with the results from the coarser resolutions (7 km and 2.8 km) that are provided in the Supplementary Material (SM).

Nudging GPS data induces a reduction of precipitation, most strongly over the western slope of the Alps and Lugano, decreasing precipitation from 50 mm to 15 mm, and over the Udine region with a reduction of from 50 to 10 mm (Fig. 4a). No shifting of the location of maxima occurs as no dynamic impacts like changes in the wind direction and intensity seem to
be introduced by the GPS nudging (not shown). Nudging RAD observations brings an increase in precipitation, both in intensity and extension (Fig. 4b). The areas most affected are located to the east of the Rhone valley, over Lugano, and Udine with up to 150 mm, well above the 50 mm simulated in CTRL-500. This impact is especially pronounced in the RAD-2.8 simulation (Fig. S2b of the SM). Nudging HR soundings brought a diverse impact for the different resolutions but overall, the HR simulations show a marked decrease of precipitation amount over Lugano (Alps) and Udine, compared to CTRL.
Intensities over these two spots are as low as 10 mm in the HR-500 simulation (Fig. 4c). However, for HR-2.8 there is an increased precipitation maximum over FR in HR-2.8, reaching values of over 100 mmd$^{-1}$ (Fig. S2c in the SM). Finally, combining all observation types for nudging (GPS-RAD-HR-500, Fig 4d) yields a structure similar of that of the RAD simulations but with a weaker precipitation increase (Fig. 4b).

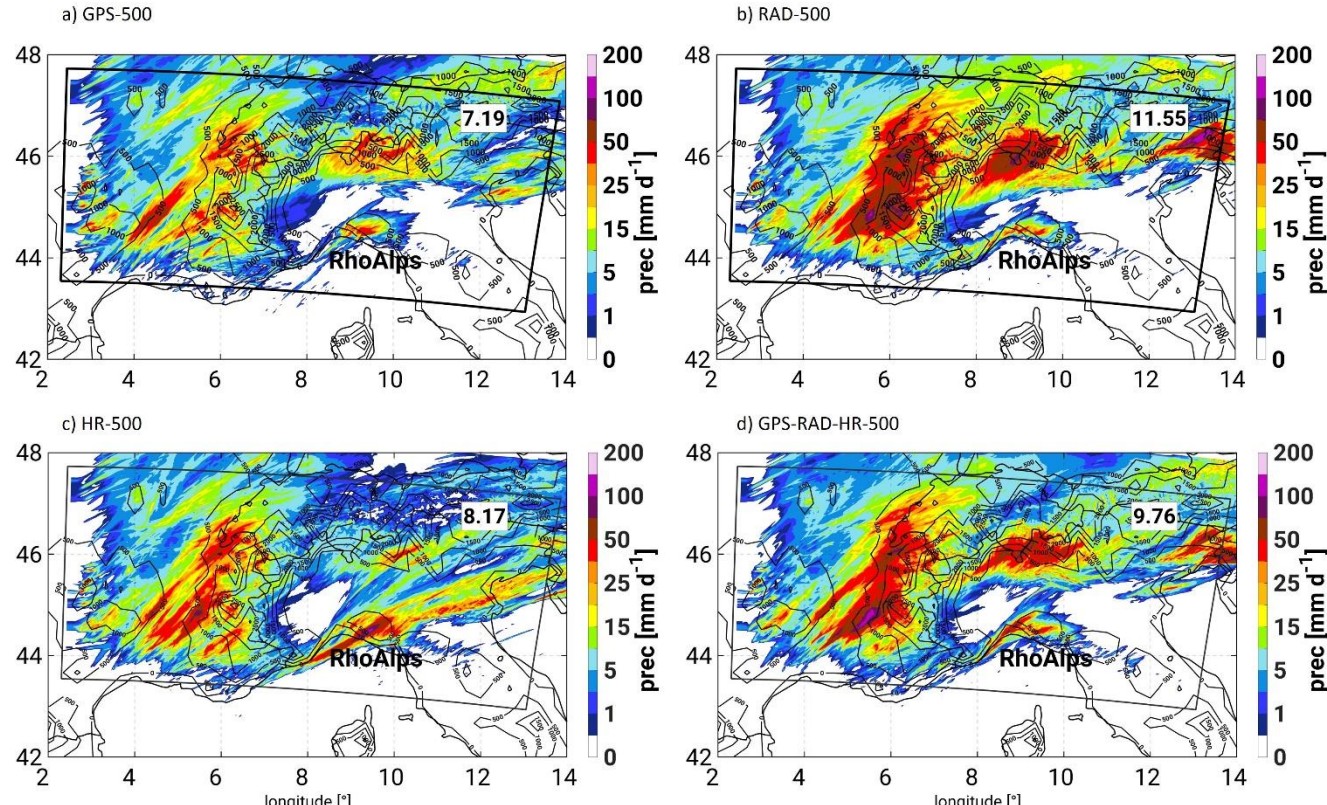

**Figure 4. As Fig. 3 but for GPS-500 (a), RAD-500 (b), HR-500 (c) and GPS-RAD-HR-500 (d). The analogous analyses using the 7 km and 2.8 km grids are shown in the SM.**



In the following we validate the assessed impact of the different observation types qualitatively by comparing precipitation observations (rain gauges and MSWEP) against the COSMO simulations. The use of MSWP (resolution of 0.1°) is motivated by the fact that it is a gridded product needed for the validation of precipitation correlation and structure. We validate for 24

September 2012, the spatially averaged 24-hourly aggregated precipitation (Fig. 5a and labels within the area RhoAlps in Figs. 3 and 4), the 99th percentile of 3-hourly aggregated rainfall (Fig. 5b) and the FSS (Fig. 5c).

Figure 5a confirms that all reference runs underestimate precipitation amount by about 4 mm. CTRL-7 shows the best result, since CTRL-2.8 and CTRL-500 emphasize more localized precipitation structures, which after spatial averaging contribute less to the final total. The simulations with nudged GPS data, further reduce the precipitation amount, worsening the values in

the comparison against observations for all resolutions with averages of ca. 8 mm only. The sole simulation able to increase the precipitation amount sufficiently is RAD with values between 16 and 12 mm, with the best representation given by RAD-2.8. This, as seen in Fig. 3b and S1b and S2b of the SM is due to larger precipitation over the western Alps and Switzerland. Nudging HR, similarly to GPS reduces the 24-hly precipitation amount resulting in worse scores for this metric. Finally, the combination of several observation types brings counteracting effects in that the drying induced by GPS and HR is counteracted

by the precipitation increase in RAD. Also noteworthy is the fact that for GPS, GPS-RAD, RAD-HR and GPS-RAD-HR the most suitable resolution is 2.8km.

Figure 5b evidences that  MSWEP cannot sample 3-hourly extreme intensities as large as those of RG, probably due to the blending of different precipitation products (CMORPH, PERSIANN, CPC, etc.), hence the large differences between the two products and the usefulness of considering both observational data sets for the validation. The comparison of the CTRL runs

shows similar intensities for CTRL-7 and CTRL-2.8 and somewhat lower (by ~5 mm) for CTRL-500. A plausible explanation is the use of a 3D closure for the turbulence scheme (see Table 1). Verrelle et al. (2015) showed that a 3D closure for the turbulence scheme induces larger horizontal diffusion in the area of the cloud base reducing convective intensity. GPS shows too weak extreme precipitation intensities for all resolutions, with the best results for GPS-2.8 (20 mm) for 3-hourly aggregations. On the contrary, RAD shows a good improvement with 3-hly precipitation intensities in the order of 27 mm,

similarly to RG, for all used resolutions. The GPS and HR simulations show too weak precipitation (between 16 mm and 23 mm) compared to RG, analogously to the underestimation of 24h sums showed in Fig. 4a and 4c. Regarding the combined observations (GPS-RAD-HR), the use of RAD increases the precipitation intensities to more realistic values. For all used observations the 2.8 km grid provides the best results compared to RG.

Finally, the FSS analysis (Fig. 5c) shows a good performance of the CTRL runs ($FSS \approx 0.91$). Nudging GPS reduces the FSS

score due to the excessive precipitation reduction, which is consistent for all resolutions. RAD improves the representation of the precipitation structure ($FSS \approx 0.95$) due to the wider rain areas over Switzerland and the Rhone valley and the eastward shift to the western side of the Alps (Figs. 4b, S1b and S2b) HR also shows no added value for the improvement of precipitation area. Combined observations (GPS-RAD-HR), show little scale dependency and an improvement for the structure thanks to the impact of RAD. For this metric the 2.8 km grid shows the best value.



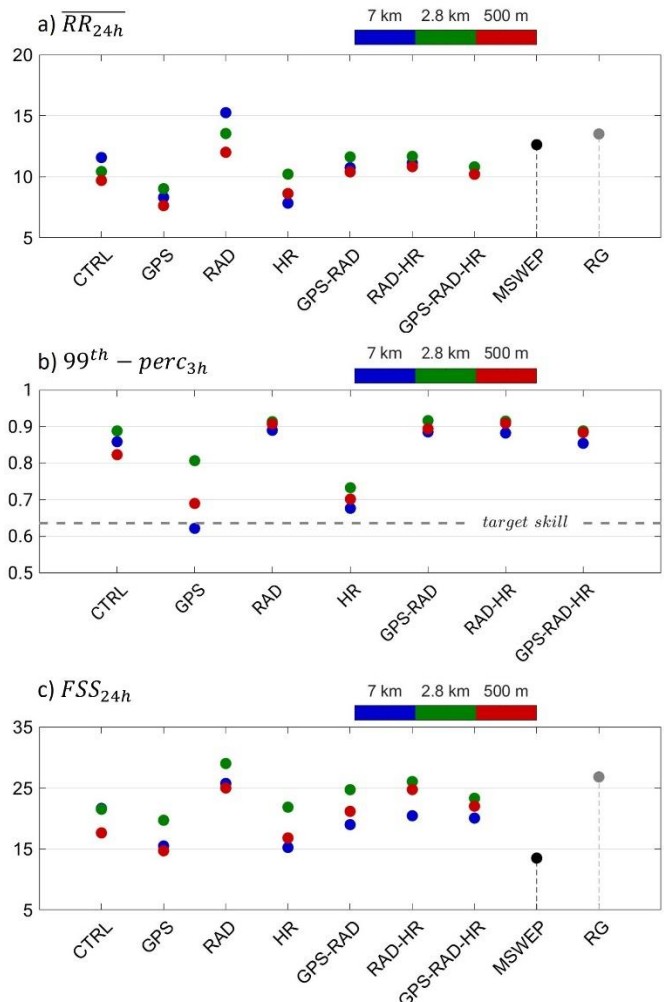

**Figure 5: a) Accumulated precipitation amount during 24 September 2012, spatially averaged over RhoAlps, for COSMO, RG and MSWEP. (b) 99th percentile in mm/3h over the investigation area RhoAlps during 24 September 2012. (c) FSS of 24h precipitation amount.**

To understand the temporal evolution of the event, Fig. 6a shows spatially averaged precipitation over the investigation area RhoAlps. Precipitation as measured by MSWEP starts on 23 September at 2100 UTC (black line in Fig. 6a), over the western part of the RhoAlps domain, reaching a steady maximum of 3 mm h$^{-1}$ in the spatial averages between 0600 UTC and 1800 UTC of 24 September (MSWEP; black colour). The convective decay is effective after 1800 UTC where the last cells over the Italian-Slovenian border start to reduce their intensity. All simulations (colour lines in Fig. 6a) capture the event with a good representation of its initiation hour, however simulating its decay between 3h and 2h earlier. As in Fig. 4, only the results for the 500m resolution are shown. The analogous results for the 7 km and 2.8 km grid are given in the SM, which show a similar response to the different observation types. CTRL-500 (blue), GPS-500 (red) and HR-500 (green) show for most of the event's duration a spatially averaged intensity lower than MSWEP (between 0.5 mm h$^{-1}$ and 1 mm h$^{-1}$), explaining the differences in





the 24-hourly aggregations (Figs. 4 and 5). Only the simulations including the operational soundings, RAD-500 (yellow), and GPS-HR-RAD-500 (purple), show a precipitation increase in agreement with the spatial distributions (Figs. 4b, 4d and 5). The

temporal evolution shows that precipitation increase occurs after 24 September 0600 UTC and reaches 4.5 mm h$^{-1}$ for the former and 3 mm h$^{-1}$ for the latter. In Section 4.3 we analyse the causes of the vast moisture increase in RAD.

To provide a quantitative score of the agreement in the temporal evolution of precipitation between observations (MSWEP; black line in Fig. 6a and the simulations; coloured lines), Fig. 6b shows the anomaly correlation for the spatially averaged 3-hourly aggregations (as presented in Figs. 6a, S3a and S4a). CTRL-7 performs better than CTRL-2.8 and CTRL-500 with a

correlation of up to 0.9 against MSWEP, due to a better location of precipitation variations at each grid point related to its more similar resolution to that of the observations (~ 11 km). Nudging GPS data improves the temporal representation of precipitation of COSMO for all grid types (Fig. 6b, S3a and S4b). This is related to a smoother representation of the precipitation increase between 0300 and 0600 UTC and a flatter curve in contrast to other observation types (Fig. 6a, S3a and S4a). This is possibly due to the ability of the GPS nudging to improve the representation of the arrival of moisture and

consequent increase, associated with precipitation initiation. RAD and HR bring little improvement, with even some deterioration for RAD-7. This is due to the quick and large precipitation increase occurring around 0500 UTC. HR also brings some improvement due to a good representation of the timing of convective decay. Combining the different observation types (GPS-RAD-HR) brings a mixed impact (improvement by GPS and HR, worsening by RAD), which conceals the dependency on the used model resolution ($r = 0.82$).

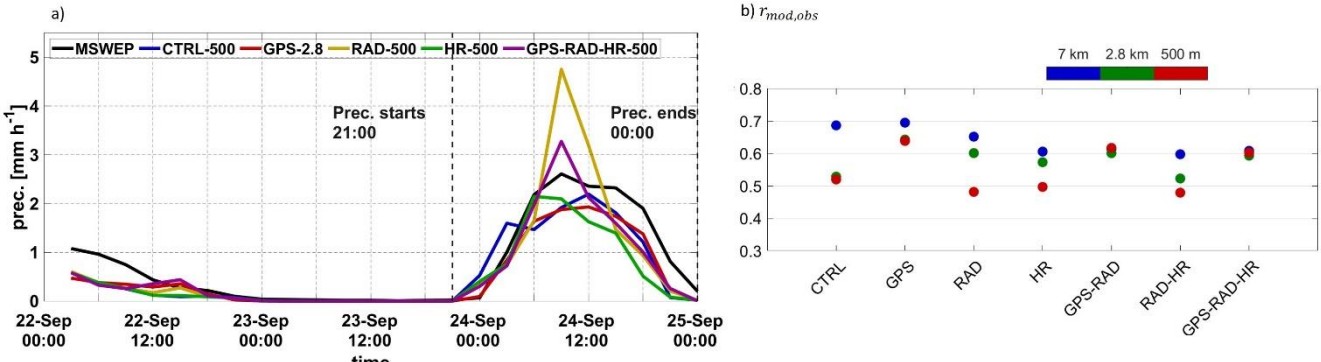


**Figure 6: Temporal evolution of spatially averaged precipitation (a) and anomaly correlation validation of the precipitation temporal evolution (b). All simulations have been coarse-grained to the MSWEP spatial resolution (0.1°). Spatial averages are performed for 3-hourly data. The corresponding results of a) for the 7 km and 2.8 km simulations can be found in Figs. S3a and S4a of the SM.**

We conclude from the previous analysis that (a) Only RAD brings an improvement to the simulation of precipitation, (b) GPS

and HR excessively reduce the simulated precipitation, which could be related to model errors in COSMO, (c) GPS brings added value in simulating the timing of the event and (d) there is overall little dependency on model resolution





## 4.2 Atmospheric Moisture

Large moisture amounts were advected with the southwesterly to southeasterly flow up the Rhone valley during 23 September 2012. The arrival of the cold front from the west initiated the HPE at 2100 UTC of 23 September, which then lasted until the

evening of 24 September. Figure 7 shows the evolution of spatially averaged hourly IWV from the 500m COSMO simulations and GPS. For the following assessment we applied a correction to IWV for height differences between the model surface and the station altitude following Bock (2005) and Parracho (2018). The correction is based on an empirical linear relationship between IWV biases and height differences (dh) following the equation $dIWV/IWV = -4 \cdot 10^{-4} \cdot dh$. The results for 2.8 km and 7 km can be found in the SM. The highest GPS-IWV amount (27 mm; black line in Fig. 7) persists for 12 hours over the

study region starting on 24 September at midnight, right after a large humidity increase associated with the arrival of the cold front and triggering of precipitation. CTRL-500 (blue line in Fig.7) reproduces the IWV temporal evolution fairly well until 1000 UTC on 23 September, when a period of considerable overestimation (+ 2 mm) begins, lasting until 0500 UTC, well after convective precipitation had started. After 1000 UTC CTRL-500 matches better with the GPS-IWV observations. An overestimation of IWV by COSMO had already been assessed by previous studies (Caldas-Alvarez and Khodayar, 2020) and

was also shown for the non-hydrostatic model AROME in Bastin et al. (2019). Nudging GPS (red line in Fig. 7 and Figs S3b and S4b of the SM) reduces the IWV overestimation until 0600 UTC on 24 September. This observation type brings the best agreement with observations throughout the complete event. This is as expected, provided that the GPS-IWV observations are not independent from the assimilated GPS-ZTDs. Nudging RAD (yellow) also corrects the IWV overestimation until 0500 UTC 24 September. However, an abrupt IWV increase takes place after 0500 UTC 24 September, with differences up to 2 mm

against observations lasting for about 5 hours. Nudging HR (green) corrects the IWV overestimation until the beginning of the event (2100 UTC 23 September), however drying excessively the investigation area until 24 September 1800 UTC. Nudging all observation types together (GPS-RAD-HR; purple) corrects the IWV overestimation until 2100 UTC on 23 September 2012 (purple line in Figs. 7, S3b and S4b).

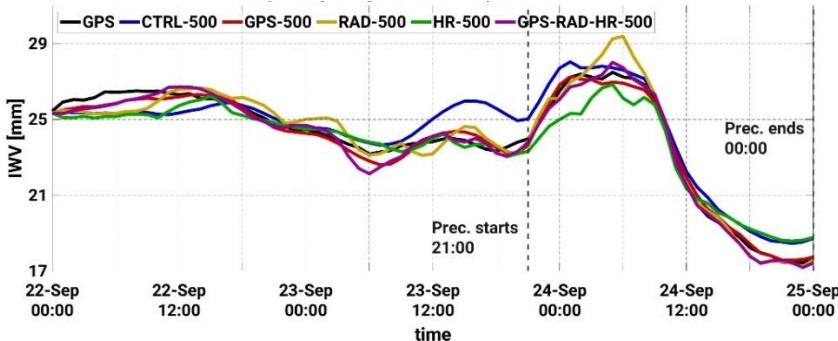

**Figure 7: Temporal evolution of spatially averaged IWV (b) for the simulations with the 500m grid. IWV is obtained through interpolation to the location of the GPS stations, applying a height correction following Bock et al. (2005) and Parracho et al., (2008). The corresponding results for the 7 km and 2.8 km simulations can be found in Figs. S3b and S4b of the SM.**





The temporal evolution in Fig.7 has shown a) the correction of the moisture overestimation by all observation types and b) the relationship between IWV fluctuations and the timing of heavy precipitation over the RhoAlps area. To provide a quantitative

assessment of the moisture representation in COSMO for this event, Table 2 shows the time averaged RMSE (left) and MB (right) between the COSMO simulations and the GPS measurements at the station locations. The MB is obtained as the MOD-OBS differences (Sect. 2.4). The results show an average RMSE between 0.79 and 1.11 mm for the CTRL runs (depending on the model resolution) with an MB between 0.18 and 0.52 mm. As expected, assimilating the GPS observations reduces these differences down to ca. 0.3 mm for RMSE and to 0.5 mm for MB. Nudging RAD shows a reduction of RMSE and MB in the

convection permitting grids (2.8 km and 500m) not seen for 7 km (increase of RMSE from 0.79 mm in CTRL-7 to 0.94 mm in RAD-7 and to 0.88 mm in HR-7). Finally, the corrections induced in GPS-RAD-HR, dominated by the influence of the GPS measurements, provide added value with changes in MB between 0.06 mm and -0.14 mm depending on the model resolution.

**Table 2. Root Mean Square Error (RMSE; left) and Mean Bias (MB; right) of spatially and temporally averaged IWV between GPS**
**and COSMO (22 September to 25 September 0000 UTC) over RhoAlps. The averages are obtained from hourly IWV values, at the GPS station locations. The COSMO simulations have been coarse-grained to a common grid of 8 km grid spacing for this comparison. And a height correction on model data based on Bock (2005) and Parracho (2008) has been applied.**

| RMSE [mm] \| MB [mm] | CTRL | GPS | RAD | HR | GPS-RAD-HR |
|---|---|---|---|---|---|
| 7 km | 0.79 \| 0.18 | 0.27 \| 0.05 | 0.94 \| -0.46 | 0.88 \| 0 | 0.39 \| 0.06 |
| 2.8 km | 1.11 \| 0.52 | 0.27 \| 0.02 | 0.87 \| -0.46 | 0.71 \| -0.06 | 0.37 \| -0.01 |
| 500m | 0.94 \| 0.4 | 0.33 \| -0.1 | 0.63 \| 0.09 | 0.8 \| -0.25 | 0.39 \| -0.14 |

The fact that the GPS and HR observations improve the IWV representation, but generate too little precipitation, is indicative
of errors in the numerics and physics for this case study. The results suggest that COSMO struggles to turn its excessive moisture content into precipitation, thus leaving the atmosphere too humid. Correcting the latter by additional observations improves IWV, drying the convective environment, however at the expense of further deteriorating rainfall.

To understand how IWV errors are distributed in the vertical profile, Fig. 8 shows the MB (straight lines) and RMSE (dashed lines) of specific humidity between COSMO and four operational radiosondes of the RAD data set (Nimes, Milano, San Pietro,
and Udine). All four stations are in the lowlands (height < 100m) to avoid biases due to surface height differences. Although this comparison is not done against an independent data set, it provides valuable information of the vertical levels at which the nudging of the different observations has the largest impact. Furthermore, given that both the operational and the special high-resolution HyMeX radiosondes were used in the nudging experiments, no other vertical humidity profiles with high-accuracy were available for an independent comparison during this period.

Regarding the 7 km resolution (Fig. 8a), CTRL-7 (blue) shows an RMSE between 0.7 g kg$^{-1}$ and 1 g kg$^{-1}$ with the largest deviations occurring at 700 hPa. These errors, however, seem to be compensating each other, since the MB lays within acceptable values (between -0.2 g kg$^{-1}$ and 0.2 g kg$^{-1}$). Similar RMSE and MB are found for GPS-7 (red) and RAD-7 (yellow),





however the latter with a slightly negative MB (ca. -0.2 g kg$^{-1}$) throughout the complete profile, indicating a drier model at the selected four low-height stations. HR-7 (green) shows the largest deviations both for RMSE (up to 1.2 g kg$^{-1}$ at 850 hPa) and

MB with an overestimation below 950 hPa and an underestimation above. This indicates that the HR soundings sampled more humidity close to the surface than the four low-height RAD stations. GPS-RAD-HR-7 (purple) has the best MB and lowest RMSE (ca. 0.6 g kg$^{-1}$) demonstrating the added value of combining these observation types. The 2.8 km resolution (Fig. 8b) shows a somewhat different vertical distribution of specific humidity for CTRL-2.8 (blue) with an overestimation of the MB between 800 hPa and 600 hPa up to 0.3 g kg$^{-1}$. RMSE values for CTRL-2.8 (blue; Fig. 8b) are high both at 700 hPa and at 925

hPa close to 1.2 g kg$^{-1}$. GPS-2.8 (red) shows similar values of the RMSE, compared to its CTRL counterpart. The vertical gradient of MB is similar to CTRL-2.8 although somewhat drier in GPS-2.8 in agreement with the IWV reduction assessed in Fig. 7 and Tab.2. This leads to the largest MB to be found in the PBL (ca. -0.2 g kg$^{-1}$) in GPS-2.8 (Fig. 8b; red). RAD-2.8 (yellow) has very good MB (0.2 g kg$^{-1}$) and RMSE values (0.8 g kg$^{-1}$), as expected, given the dependence on the observations in this comparison. This influence can also be seen for the good scores of GPS-RAD-HR-2.8 (purple). For the 500m resolution

(Fig.8c), CTRL-500 (blue) shows an underestimation of moisture in the PBL and an overestimation above 800 hPa, up to 0.2 g kg$^{-1}$. The impact of the different observation types is analogous to that observed in the 2.8 km simulations with the exception of HR-500 (green; Fig. 8c), namely, GPS-500 (red) dries the atmospheric profile, RAD-500 (yellow) and GPS-RAD-HR-500 (purple) show the best agreement with the observations and HR-500 (green) presents a slight negative bias of -0.3 g kg$^{-1}$.

The previous assessment leads to the conclusions that a) COSMO misrepresents the humidity vertical gradient for this case

study (too wet between 500 hPa and 850 hPa, also found for another case study of the HyMeX period in Caldas-Alvarez and Khodayar, (2020); and b) nudging GPS did not help improve the representation of the vertical humidity gradient, as the correction at each level is applied based on the first guess. The latter explanations imply that COSMO should have simulated stronger convective updrafts to generate more precipitation at the surface and larger transport of moisture from the PBL to the LFT in the CTRL and GPS simulations.






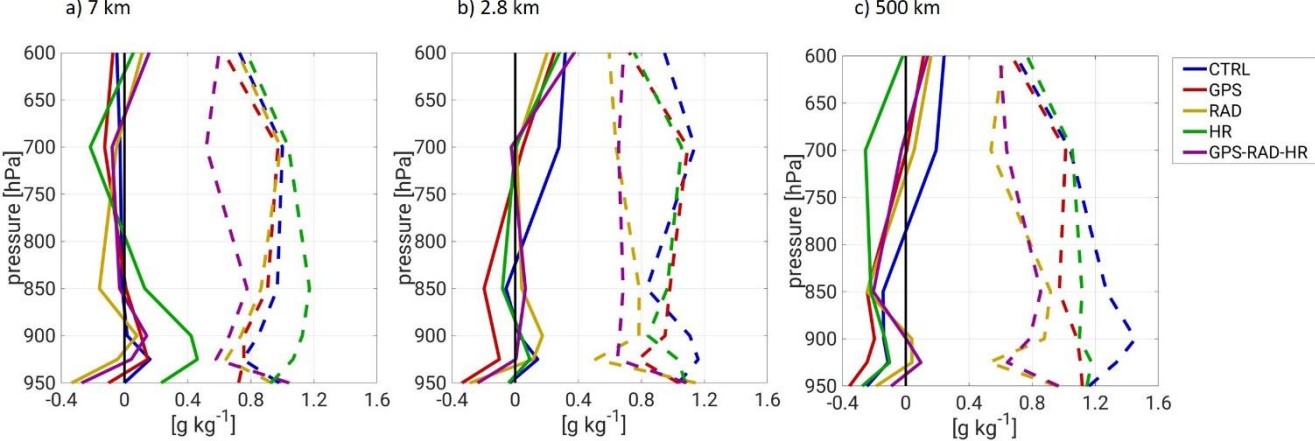

**Figure 8: Mean Biases (solid lines) and RMSE (dashed lines) of specific humidity between the operational soundings for (a) the 7 km, (b) 2.8 km and (c) and 500 m simulations. The differences are obtained at the four stations within the investigation area RhoAlps (Milano, Nimes, Udine, San Pietro) and are averaged for all stations and for the complete simulation period (0000 UTC 22 September to 0000 UTC 25 September 2012).**

### 4.3 The relevance of the Nimes 0500 UTC sounding

The good scores shown by RAD in the precipitation evaluation (Fig. 5) and the large increase of IWV and precipitation are worth an in-depth analysis of the impact of RAD on the humidity distribution and convective processes responsible for the remarkable precipitation increase. Figure 9 summarizes relevant information about the impact of RAD on humidity and precipitation. Between 23 September and 24 September, before the arrival of the cold front, vast moisture amounts were transported up the Rhone valley by the southwesterly circulation (arrow in Fig. 9a). The moisture gathering up the valley preconditions the HPE. Once precipitation starts, the Nimes RAD sounding at 0515 UTC (hereafter referred to as Nimes_0515) measured 6.5 g kg$^{-1}$ of specific humidity at 700 hPa (Fig. 9b). Compared to other soundings (either operational or high-resolution) released in the area (Fig. 9b), Nimes_0515 measured between 1.5 g kg$^{-1}$ and 2.5 g kg$^{-1}$ more specific humidity. For example, over Candillargues at 0314 UTC on 24 September specific humidity at 700 hPa was 5 g kg$^{-1}$ and over Marseille at 0555 UTC it was lower than 4 g kg$^{-1}$. This implies that after its assimilation, specific humidity at that level was considerably increased due to this one particular sounding. To demonstrate this aspect Fig. 9c shows that the reference runs of COSMO (CTRL-7 in blue) at that time over Nimes have a 700 hPa level 1.5 g kg$^{-1}$ drier than the observation. Hence the correction of humidity at that level after 0515 UTC is crucial for the precipitation increase observed for RAD.

The Gaussian horizontal spreading of information induced by the Nudging scheme (Schraff and Hess, 2012) and the transport of humidity with the south-westerly mean flow causes much wetter mid-levels over the Rhone valley and over the western Alps. This impact was similar for all resolutions. To demonstrate quantitatively this impact, Fig. 9d shows relative precipitation differences in % between the RAD-7 simulation and an auxiliary RAD-7 simulation where the Nimes_0515 sounding is dismissed. The results show that the contribution to precipitation of the Nimes_0515 sounding is a 40% increase spatially averaged over the whole domain and up to 70% downstream of Nimes.



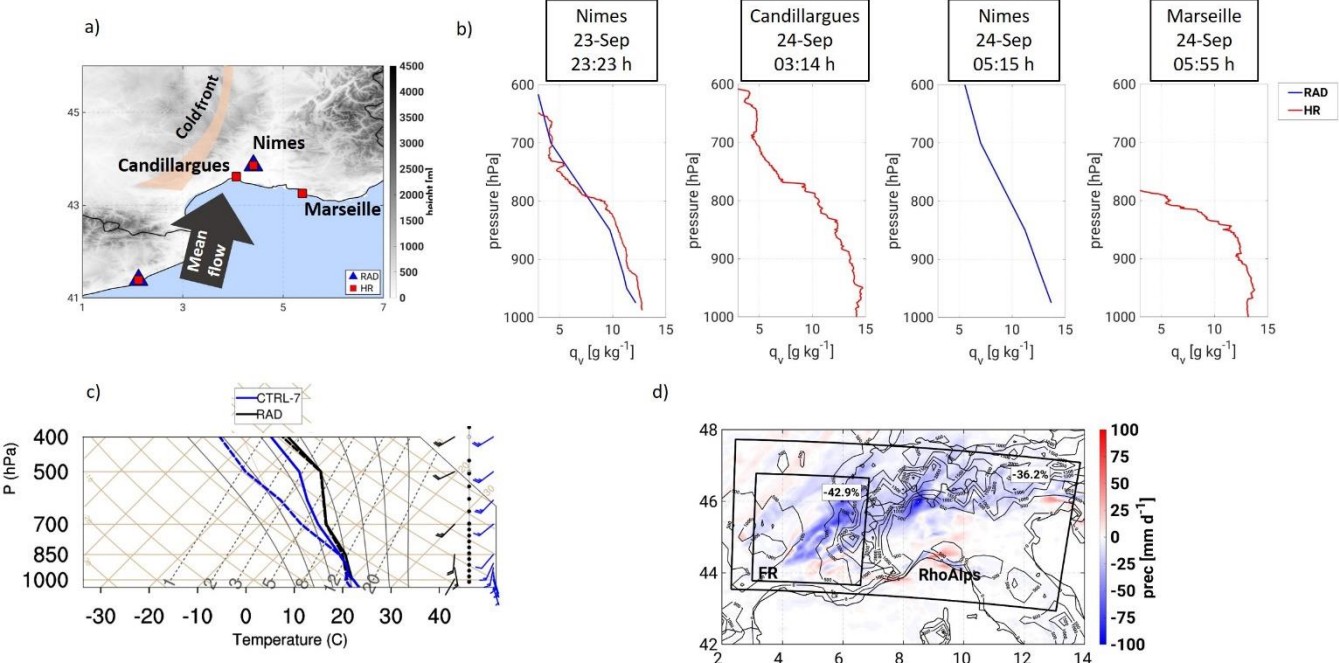

**Figure 9. (a) Location of the stations with RAD and HR profiles between 23 September 2323 UTC and 24 September 0555 UTC, as well as cold front position and direction of the mean flow. (b) Radiosonde measurements (RAD and HR) in the area. (c) Skew-T log p diagram of the Nimes radiosonde at 0515 UTC on 24 September and the simulation of the profile by CTRL-500, interpolated to the station location. (d) Precipitation differences between RAD-7 and the same simulation without the Nimes sounding shown in (c).**

The large impact of the Nimes_0515 sounding gives important clues as to whether GPS systems were able to compensate radiosondes for this case study. With no means to measure the vertical distribution of humidity, GPS struggles to bring the expected improvement in precipitation representation. The reason why other soundings close to the Nimes in time and space did not measure such a large humidity amount at 700 hPa is still unknown. Unfortunately, no other humidity observations exist for that time and location (LIDAR, pressurized balloons or dropsondes). A possible explanation is the embedding of the Nimes_0515 sounding within a precipitating system. In that case a saturated atmosphere would be present at 700 hPa exactly where the sounding was launched. This implies that in the RAD simulations the existing errors in COSMO regarding the underestimation of humidity at the LFT (see Fig. 8) and the need of excessive moisture to represent sufficient rain are compensated by this one sounding. This highlights the relevance and complications of targeted observations for DA. Even in a field campaign especially dedicated for precipitation studies, spatial distances of 60 km and temporal differences of 30 minutes are enough to miss a crucial measurement of water vapour. This demonstrates that very frequent sampling over a large areas is needed in some cases to monitor water vapour distribution for assimilation.



### 4.3.1 Impact on moisture flux, instability, and wind circulation

In order to better understand the precipitation increase due to nudging, we now investigate its impact on moisture advection, temperature, and instability. We focus on the 700 hPa level due to the humidity differences assessed earlier in this section. The temporal period analysed are the 6 hours following the large precipitation increase in RAD and the investigation area is now

FR (Fig. 1), where the largest impact of the Nimes_0515 sounding was seen. Only the 500m results are shown given the analogous impact in the other two resolutions, the results for the 7 km and 2.8 km grid can be found in the SM.

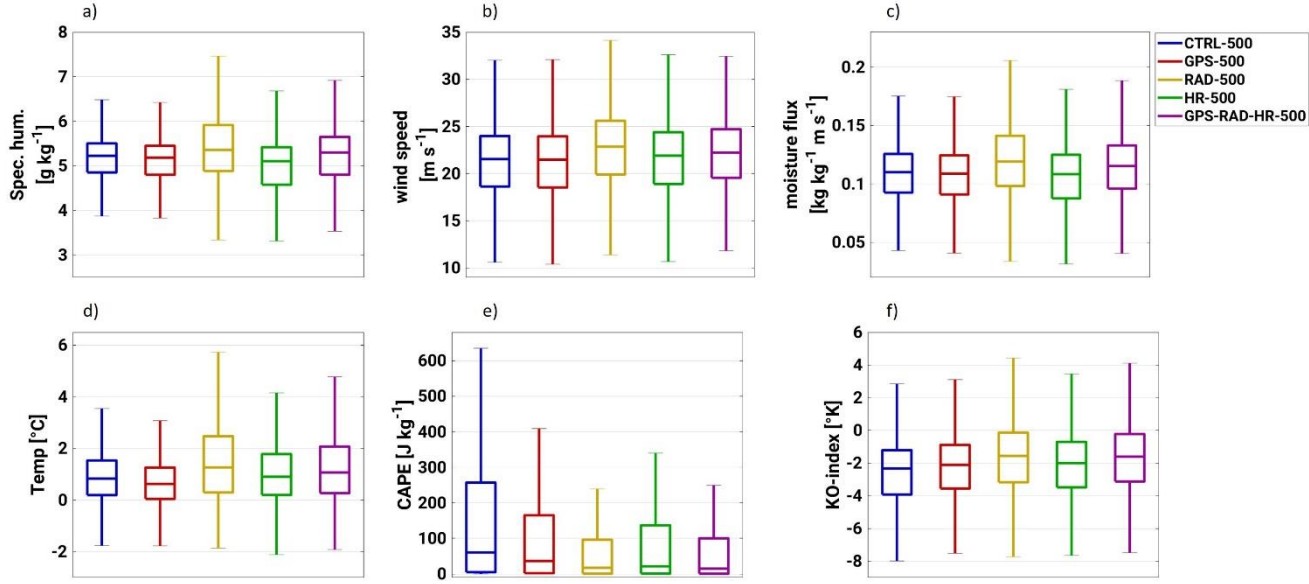

**Figure 10: Box-and-whisker-plots showing the median, quartiles and extremes of specific humidity (a), wind speed (b), moisture flux**
**(c), temperature (d), CAPE (e) and KO-index (f) at 700 hPa. All the values have been obtained from hourly COSMO output between 0500 UTC and 1000 UTC of 24 September 2012 over the study region FR.**

Figure 10a shows that GPS-500 (red) and CTRL-500 (blue) have a similar distribution of specific humidity at 700 hPa, with median values of 5.3 g kg$^{-1}$ and extremes as large as 6.5 g kg$^{-1}$. The impact of exclusively nudging RAD (yellow) soundings is an increase of specific humidity to 5.5 gkg$^{-1}$ in the median and 7.4 gkg$^{-1}$ for the 75th percentile. This is mostly due to the

influence of the Nimes_0500UTC sounding as discussed above. The impact of HR (Fig. 10.a), in contrast, is a reduction of the median and larger variability compared to CTRL-500. The GPS-RAD-HR-500 (purple) simulation shows increased humidity, mostly due to the RAD contribution. Regarding horizontal wind speed (Fig. 10b), GPS-500 (red), HR-500 (green), and GPS-RAD-HR-500 (purple) show hardly any differences compared to CTRL (blue). However, the Nimes_0515 sounding enhances the speed at this level, probably due to enhanced convection inducing stronger winds. The impact for moisture flux

at 700 hPa (Fig. 10c) can be understood as the combination of humidity and wind changes. CTRL-500 (blue), GPS-500 (red), and HR-500 (green) show very similar median and extreme values of moisture flux, with 0.12 kg ms$^{-1}$ for the former and 0.18 kg ms$^{-1}$ for the latter. For their part, RAD-500 (yellow) and GPS-RAD-HR-500 (purple) show an increased moisture flux with





extreme values reaching 0.2 and 0.19 kg ms$^{-1}$, respectively (Fig.10c). Regarding temperature (Fig. 10.d), GPS-500 (red) slightly reduces the values at 700 hPa, likewise specific humidity, due to weaker convection less latent heat is released in the

process of condensation and less mixing occurs from the PBL to the free troposphere. However, the RAD-500 (yellow) simulation shows a large variability and larger extremes with values 2°C higher. This further supports the hypothesis that the Nimes_0515 sounding sampled in a precipitation area affected by latent heating and vertical moisture fluxes. Finally, GPS-RAD-HR-500 (purple) shows 75th percentile values up to 4.5°C influenced by the RAD and HR measurements. Regarding atmospheric instability, Fig. 10.d represents Convective Available Potential Energy (CAPE; Moncrieff and Miller, 1976), and

Fig. 10.e KO-index (Andersson et al., 1989). CAPE provides a quantitative estimation of the energy available for lifting of a hypothetical air parcel in the lowest 50 hPa of the atmosphere. The KO-index provides an estimation of potential instability expressed as differences in equivalent potential temperature ($\theta_e^{850}$) between 500 hPa and 850 and 700 hPa and 1000 hPa. CTRL-500 (blue) shows the largest atmospheric instability (high CAPE, low KO-index). The nudged simulations show lower instability (CAPE and KO-index). In the case of GPS-500 (red) and HR-500 (green) explicable from the drying of the

atmospheric profile down to the surface, which reduces equivalent potential temperature ($\theta_e$) for both CAPE and KO-index calculations. For RAD-500 (yellow) and GPS-RAD-HR-500 (purple), the moisture increase at 700 hPa is interpreted as an increase of $\theta_e$ at that level, hence leading to a less steep lapse rate decreasing CAPE (Figs. 10e, S5e and S6e) and increasing KO-index. It is worth noting that for this case study, not only the low-level conditional instability defines the environment for convection but also the cold front and upper-level divergence that release potential instability. From this analysis we conclude

that after 0500 UTC the humidity increase at 700 hPa was the dominating factor invigorating convection.

## 5 Conclusions

This study assessed the impact of nudging GPS column water vapour estimates, operational soundings, and high-resolution soundings on high-resolution model simulations using an autumn convective precipitation event in the western Mediterranean as a case study (HyMeX-IOP6). The high density of observations obtained in the framework of HyMeX allowed a thorough

investigation of assimilation experiments to systematically assess the added value, advantages and disadvantages of the individual observation types and the sensitivity to model resolution. For example GPS lacks vertical information but has a vast coverage in the western Mediterranean and a temporal resolution of ten minutes, whereas high resolution radiosondes have a high vertical resolution (~ 700 levels) but a scarce coverage and sub-daily temporal resolution (6h to 12h). We performed the sensitivity experiments using the COSMO model and the Nudging scheme in model resolutions of 7 km, 2.8 km, and 500m.

The main conclusions are:

    a)   COSMO shows deficiencies in representing the mechanisms of heavy precipitation for this case study, which could not be corrected by nudging additional observations. The reference runs (no assimilated data) showed a moist bias before precipitation onset and an underestimation of precipitation, indicating that COSMO is unable to transform the





excess of moisture (especially at the mid-levels into precipitation). Nudging GPS and HR data corrected this moist
bias but also further reduced precipitation, leading to worse verification scores irrespective of resolution.

b)    Nudging operational radiosondes, however, brought a clear improvement in the representation of 24-h precipitation,
precipitation intensities and spatial structure. The improvement was brought about by a large precipitation increase
(+ 40% in the 7 km simulations) after 0500 UTC on 24 September lasting 3 hours. This was mainly caused by the
assimilation of one particular sounding in southern France (i.e. from Nimes at 0515 UTC on 24 September), probably
embedded in a precipitating convective cell, south of the main convective systems. The main mechanism was an
increase of specific humidity of 2.5 gkg$^{-1}$ at the 700 hPa level, 5 hours after precipitation initiation, which likely
reduced the entrainment of dry air and led to higher moisture availability.

c)    The large impact brought about by an individual sounding implies, on the one hand, that traditional sounding systems,
which need manned operations and have a lower spatial coverage and temporal resolution will still be needed, even
when GPS networks are also available. This is further supported by the difficulties of GPS observations to correct the
vertical distribution of specific humidity. On the other hand, it implies that targeted observations, such as the ones
carried out in HyMeX can in fact be decisive for assimilation in convective situations. Especially, for variables with
large spatial and temporal variability such as atmospheric moisture.

d)    The overall performance and type of impact of each observation type were not dependent on the used model
resolution. The 2.8 km resolution showed marginally better precipitation scores for all used observations suggesting
that a computationally more expensive resolution of 500m is not needed for this case study. As 2.8 km is the
operational configuration, model parameters are optimally set for this resolution possibly giving in an advantage that
could be eliminated with a re-tuning for 500m.

e)    We would like to highlight the added value of GPS nudging in improving the temporal evolution of precipitation.
GPS improves the anomaly correlation for all resolutions suggesting that nudging together GPS and soundings can
benefit both from the temporal evolution improvement and the vertical resolution of the radiosondes.

The fact that COSMO underestimates the precipitation amount with a too moist pre-convective environment in this case points
to model errors in the physical parameterizations or numerics, which assimilation procedures could not compensate. In a
follow-up study we investigate whether physics updates in the framework of the development of the successor model ICON
have been able to reduce this problem. In addition, more cases need to be investigated to generalize the findings presented here
as planned for a follow-up study covering the entire autumn of 2012.

## 6 Code availability

The COSMO model is only accessible after request to the consortium and after agreeing on the terms and licences at
http://www.cosmo-model.org/. Parts of the model documentation are freely available at https://doi.org/10.1127/0941-
625    2948/2008/0309.

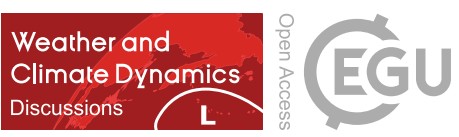

## 7 Data availability

The simulation data for precipitation and IWV used to produce Figs. 3b-d, 4, 5, 6, 7, 9d, S1, S2, S3, S4, and Tab. 2, as well as post-processed humidity, instability, temperature, and wind model data used for Figs 8, 9b-c, 10, S5 and S6, are accessible at https://bwdatadiss.kit.edu/review/access/06ec49a7e16a1ba080e8ee1fbcd292eee364e18807be6f583c585de1483d58e7.

Regarding observations, all data sets were provided by other groups or institutions, hence access is only possible after agreement with the corresponding authors. Most of which belong to the HyMeX/MISTRALS data repository. All observational data sets are referenced and contact details are provided when available.

## 8 Author Contribution

SK designed and planned the experiments and supervised ACA during his PhD work to which this investigation belongs. ACA

performed the nudging simulations. SK, PK and ACA analysed the results. ACA and PK wrote the manuscript with comments from SK.

## 9 Competing Interests

The authors declare that they have no conflict of interest.

## 10 Acknowledgments

This research work has been financed by the Bundesministerium für Bildung und Forschung (BMBF; German Federal 605 Ministry of Education and Research) project PREMIUM 01LN1319A.

We would like to thank the HyMeX project and MISTRALS through the grants by grants MISTRALS/HyMeX and ANR-11-BS56- 0005 IODA-MED for providing most of the observational data used for assimilation and model validation. These are the RG, the operational soundings, the high-resolution soundings and the GPS-ZTD and GPS-IWV products. The latter two

data sets have been contributed by Oliver Bock and the LAboratoire de Recherche En Géodésie (LAREG) of the French Institute of the Geographic and Forest Information (IGN).

We also want to thank Hylke Beck for sharing the MSWEP precipitation data for the study period (Beck et al., 2017).

Regarding the COSMO model, we are grateful to the German Weather Service (DWD) for sharing a distribution of the model and in particular to U. Schättler, for supporting the installation of the model, C. Schraff for helping to install and running the

nudging scheme and U. Blahak for his scripts to run sub-kilometre simulations with COSMO.

The whole simulation effort would not have been possible without the support from the Steinbuch Centre for Computing, hosted at KIT, in particular to the high-performance Computer Forschungshochleistungs-rechners II (ForHLRII).





PK acknowledges project B6 ''New data assimilation approaches to better predict tropical convection'' of the Transregional Collaborative Research Center SFB/TRR 165 ''Waves to Weather'' funded by the German Science Foundation (DFG).





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
