# Peer review of "The impact of GPS and high-resolution radiosonde nudging on the simulation of heavy precipitation during HyMeX IOP6"

_Weather and Climate Dynamics, 2021_

## Referee Comment (RC1)

| Model configurations: | | | | |
|---|---|---|---|---|
| Resolution | Forcing | Lev. | Convec. | Turb. |
| 7 km | IFS | 40 | Tiedtke Deep | 1D TKE |
| 2.8 km | CTRL-7 | 50 | Tiedtke Shallow | 1D TKE |
| 500m | CTRL-2.8 | 80 | - | 3D TKE |

| Assimilation configuration: | | |
|---|---|---|
| Observations | Freq. | Levels |
| RAD (Oper. Rads.) | ~ 6 h | ~20 |
| HR (High-res. Rads.) | ~6 h | ~4000 |
| GPS | 10 min | Integr. |
| GPS-RAD | Combined instruments | |
| RAD-HR | | |
| GPS-RAD-HR | | |
| CTRL (No obs.) | | |

---

## Author Response (AR1)

**Answers to Reviewer 1, Dr. Dominik Jacques - wcd-2021-2**

We would like to thank Dr. Dominik Jacques for his valuable comments and corrections. We have accepted most of the remarks and included them in the revised version of the manuscript. In the following, we provide detailed answers to his questions/requests. In case our corrections need no further explaining, they have been directly included in the new version of the manuscript. The specific changes can be seen in the "track_changes" version.

**General comment**

*This manuscript examines forecasting experiments where radiosonde and GPS delay observations are assimilated before a significant precipitation event. The main goal being pursued is to establish whether increased model and/or observation resolution can bring significant improvements to the forecasts.*

*Various combination of model resolutions and observations are tested. The performance of these forecasts is mostly assessed from the resulting precipitation compared against observations. The overall conclusion is that the assimilation of operational radiosonde data is important but assimilating extra "high-resolution" observations is not. Deficiencies in modelled moist processes and lack of vertical information in GPS observations are given as factors that could explain the results obtained.*

*With their heterogeneous distributions and difficult statistical properties, "physical" state variables such as moisture and precipitation remain challenging to data assimilation and verification. As such, this manuscript takes place in the context of an active topic of research. While the experiments and analyses presented are not fundamentally novel, they contribute to a better understanding of data assimilation for moist processes. The topic is interesting and within the scope of the Weather and Climate Dynamics journal.*

*The manuscript is well organized and generally easy to follow. The in-depth examination of the meteorological impacts (i.e. changes in moisture) brought by the assimilation process is interesting.*

*Perhaps the area that needs the most improvement is the description of results related to figure 5. As discussed in major comment 1 below, the description of certain scores is missing or unclear. There is also a labeling error in figure 5.*

A detailed review of the changes carried out related to Fig. 5 is included later in this document in the Major Comments section.

*Only one precipitation case is presented in this study. On the one hand, this allows for an in depth analysis of the factors contributing to this precipitation event. On the other hand, this imposes a strong limitation on the generalization of conclusions drawn from the various analyses. Luck (good or bad) cannot be ruled out of the many factors influencing the forecasts. Interestingly, the analysis reveals that the assimilation of one radiosonde in the operational network has a significant impact on the forecasts being performed*

The reason for using just one case study was to be able to focus on the different impacts of each observation type. With a total number of 21 simulations, 3 observation types (and their combinations) in 3 different resolutions, considering several cases would have been challenging.

We planned these experiments as an illustrative means of assessing the improvement potential of each observation type. Indeed, these experiments belong to a series of GPS assimilation experiments reproducing the whole 2012 Autumn period, where we got further insights on the model biases, regarding water vapour and precipitation. Analysing the impact of the different observation types on

all cases of the 3-month period, would have not allowed such an in-depth assessment. This is why we simulated IOP6 separately.

Nevertheless, as pointed out, the manuscript should clearly state that the findings relate to this one case study, and that generalisation of the results is therefore constrained. The following clarifications have been added in,

The conclusions:

*"The fact that COSMO underestimates the precipitation amount with a too moist pre-convective environment in this case points to model errors in the physical parameterizations or numerics, which assimilation procedures could not compensate. The results of this case study provide a first assessment, but further cases should be analysed to allow for generalization of the findings. Moreover, in follow-up work we investigate all precipitation events of the autumn 2012 and whether physics updates in the framework of the development of the successor model ICON have been able to reduce the highlighted problems"*

And abstract:

*"Future work will aim at a generalisation of these conclusions, investigating further cases of the autumn 2012 and the Icosahedral Nonhydrostatic Model (ICON) will be investigated for this case study to assert whether its numerical and physics updates, compared to its predecessor COSMO, are able to improve the quality of the simulations."*

We believe however, that our findings are of use to other researchers in the field as information on the impact of different observation types is of high relevance to improve numerical weather predictions.

*One can wonder if the conclusions of the manuscript would have been different had this radiosonde been part of the extra "high resolution" observations being tested. The examination of only one precipitation event should not prevent the publication of this manuscript. However, the limitations that come from this should be emphasized in the concluding statements. Special care should be taken with respect to the model's treatment of moist processes (section 5a) which seem to be supported by other studies but which may only be applicable to this one case.*

We agree with the reviewer that the conclusions would have been different had the Nimes_0515 sounding be part of the high-resolution data set. This is precisely one of the main points in our manuscript. Data assimilation of heterogeneous variables that are affected by non-linear processes such as precipitation is by no means straight forward. As demonstrated in the manuscript, the assimilation of this particular sounding, obtained at the right time and place, affected dramatically the final precipitation scores. This is why we deem important further investigation in targeted observation systems that sample such variables at high-resolutions since this can make the difference between a good and a bad forecast.

This is precisely highlighted in point c) of the final conclusions

*"c) The large impact brought about by an individual sounding implies, on the one hand, that traditional sounding systems, which need manned operations and have a lower spatial coverage and temporal resolution will still be needed, even when GPS networks are also available. This is further supported by the difficulties of GPS observations to correct the vertical distribution of specific humidity. On the other hand, it implies that targeted observations, such as the ones carried out in HyMeX can in fact be*

*decisive for assimilation in convective situations. Especially, for variables with large spatial and temporal variability such as atmospheric moisture."*

And in the analysis of Fig. 9

*"This implies existing errors in COSMO regarding the underestimation of humidity at the LFT (see Fig. 8) and the need of excessive moisture and rain are compensated by this one sounding. This highlights the relevance and complications of targeted observations for DA. Moreover, it also highlights that, for this case study, the accurate location and timing of that one sounding were more relevant for precipitation simulation than the higher vertical resolution offered by the HR data set. Spatial distances of 60 km and temporal differences of 30 minutes are enough to miss/capture a crucial measurement of water vapour."*

**Major comment**
*Figure 5 is problematic as it presents results that are not consistent with the verification metrics presented in sections 2.4. Addressing this issue is important as this figure is the basis for most of the discussion later in the manuscript.*

Figure 5 has been completely reworked, together with the supporting text description, which needed, as mentioned in the reviewers' comments, more readability and suppression of redundant information. In the following the different changes and improvements of Fig.5 are described.

The most relevant improvement is the dismissal of RG for this model verification. Due to comments raised by the second reviewer we have decided to restrict the results of Fig. 5 to the verification against the MSWEP product. The reviewer raised the question how the spatial scale of RG was comparable to that of MSWEP and COSMO (after coarse-graining). Indeed the spatial scale of the RG differs from the gridded data sets due to its heterogenous coverage, where distances between RG range between ca. 5 km and 20 km, depending on the location. To study further this aspect we attempted gridding the RG product by means of Inverse Distance Weighting (IDW) interpolation to the MSWEP grid (0.1°). However the interpolation of station information to a grid is not straight forward and spurious artifacts in the final data appeared. Hence the decision to restrict the quantitative verification analysis (Fig.5) to MSWEP. In the following we include the raised question and explanation of our decisions.

> *Reviewer 2: Moreover, how are the raingauges treated? For example, for the domain average, are they aggregated to the same grid of MSWEP to take into account spatial variability?*

> *Answer: We used the values of RG 24hly and 3hly precipitation aggregates at each station to obtain the spatial average (Fig.5a) and the 99-perc percentile (Fig.5b). Hence, in the old version of the manuscript, no interpolation to a common grid (0.1° from MSWEP) was carried out. This, as pointed out, can pose problems regarding comparability between the different data sets.*

> *To further extend this analysis, we have carried out interpolations of the RG 24hly and 3hly to the 0.1° MSWEP native grid by means of an Inverse Distance Weighting Method (Hodam et al., 2017). However, the interpolation has revealed spurious artifacts around the point stations and unrealistic precipitation gradients with no agreement with the original RG distribution. See for example the unrealistic dotted features in Fig.R1a or the artificial precipitation gradients in the northwestern part of the domain in Fig.R1b*

a)

[Figure]

b)

[Figure]

*Fig R1. Spatial distribution of 24hly precipitation on 24 September 2012 a) and 3hly precipitation on 24 September 2012 at 21:00 UTC. The represented data set are the RG after interpolation through Inverse Distance Weighting (IDW) to the MSWEP grid (0.1°).*

*Therefore we have decided to restrict the verification to the MSWEP product to avoid the artefacts produced by the gridding of RG data. This decision is further supported by the good results of MSWEP in previous publications (Beck et al., 2017), the fact that it already contains station data and that it complies better with the spatial scale of the coarse-grained COSMO results.*

*For example, it is not clear what the "percentiles" presented in this figure are or where they come from. Section 2.4 gives a good summary of the verification metrics used in the rest of the study. The percentiles appearing in figure 5 should be introduced there.*

We have added a new subsection (2.4.1) to introduce how the percentiles are calculated, within section 2.4 Verification metrics.

*"2.4.1 Percentile-99 of three hourly precipitation aggregates*

*We validate extreme precipitation intensity simulated by COSMO against MSWEP. To this end we upscale COSMO's grid to the MSWEP spatial resolution (0.1°) by means of bilinear interpolation. Then we obtain 3-hourly precipitation aggregates for the grid points within the investigation area. The 99-percentile is obtained from the sample of all 3-hourly precipitation intensities at each grid point during the day of precipitation i.e., for eight time steps during 24 September 2012."*

This sub-section is adequately referenced within the description of Fig. 5 in section 4.1.

*"How these metrics are computed is introduced in Sect. 2.4."*

*Also, there appears to be a labeling mistake for the y-axes of panel c. The Fraction Skill Score is in the range [0,1] but the y-axis of panel c) goes from 5 to 35. Because of this, most of the discussion on lines ~350-380 is difficult to follow and/or interpret. It is believed that this part of the text and figure 5 should be reworked before publication of the manuscript.*

Panels b) and c) in Fig. 5 were wrongly interchanged. This has been corrected in the new version of the manuscript and the text has been reworked. See the "track_changes" version.

*Still on the topic of verification, the use of "anomaly correlation" (section 2.4.1) for the verification of precipitation in a day-to-day forecasting context is unusual and somewhat confusing. If the concept of anomalies makes sense in a climatological context, it is more difficult to apply in a weather context. In my understanding, the "anomalies" should refer to some departure from a preferred mode for the model solution. Because the mode of high-dimensional pdfs are generally difficult to estimate, they are often replaced by the average of a large number of such solutions. Many seasons are averaged for climate forecasts, many ensemble members may be averaged for ensemble forecasts. In the present case, for a single weather event in a deterministic context it does not seem possible to know the "normal" mode about which the anomalies could be estimated. In particular, the daily average precipitation for one case cannot be thought of as "normal" baseline against which anomalies can be estimated.*

*That said, the correlation coefficient between two fields can be used in the context of verification. To avoid the confusion that arise from the concept of anomalies, it is suggested that correlations be estimated from the fields themselves. Just remove the \overbar{mod} and \overbar{obs} from eq. 5. The results previously obtained will be unchanged since the Pearsons's correlation coefficient is invariant to such offsets by constant values.*

We agree with the reviewer that treating the precipitation average of one case cannot be understood as the normal baseline of the event and changed this to a full timeseries correlations .

We have computed the correlation coefficient as suggested by the reviewer, removing the subtraction of the timely means, with an invariant result.

However, the formulation of Eq. 5 remains the same, as it is the formulation of Pearson's correlation coefficient. We have adapted the text to better explain that in Eq. 5, $obs$ and $mod$ stand for the spatially averaged precipitation for time step $t = i$ measured by MSWEP and simulated by COSMO, respectively. Without subtraction of the period mean, as suggested by the reviewer.

Section 2.4.2, hence is presented as follows:

*" 2.4.2 Temporal Correlation*

*In Section 4.1, we validate the precipitation temporal correlation of the different simulations against observations (MSWEP). To this end, we calculate the Pearson's correlation coefficient between the model's spatially averaged precipitation (mod) and that of the observations (obs, Joliffe and Stephenson, 2011) for 3-hourly aggregates during the day of precipitation (24 September 2012).*

$$r_{mod,obs} = \frac{\sum_i^{24h}(mod_i - \overline{mod})(obs_i - \overline{obs})}{\sqrt{\sum_i^{24h}(mod_i - \overline{mod})^2}\sqrt{\sum_{i=1}^{24h}(obs_i - \overline{obs})^2}} \qquad [5]$$

*For its interpretation it should be noted that the forecasting efficiency of Pearson's correlation coefficient is non-linear, i.e. small improvements of $r_{mod,obs}$ for values closer to 1 imply larger*

*forecasting efficiency than improvements of the same extent for values closer to 0 (Jacques et al., 2018).*

*The spatial averaging is performed over the investigation area RhoAlps, where only land points are considered due to the lack of data of MSWEP over the sea; all simulations are coarse-grained to the MSWEP resolution. "*

*As a final note, one should remember that due to its non-linear response, Pearson's correlation coefficient is difficult to interpret in the context of verification. This problem is discussed in the appendix of .*

A reference to this aspect and the mentioned publication, dealing with the non-linear relation between the correlation coefficient and the forecasting efficiency, has been included in Section 2.4.1.

**Minor comments**

*Table 1 summarizes the description of the different experiments performed in this study. The current titles for the panels of this table make its interpretation difficult. It is believed that small adjustments to the labeling would help. The figure "suggested_changes_to_table1.pdf" joined to this review presents suggestions for changes.*

We have included the reviewers comments and Table 1 has been changed accordingly:

| Model Configuration | | | | | | | | Assimilation Configuration | | |
|---|---|---|---|---|---|---|---|---|---|---|
| Resol. | Forcing | Lev. | Convec. | Turb. | Orogr. | Soil | | Observations | Freq. | Levels |
| 7 km | IFS | 40 | Tiedtke Deep | 1D TKE | | | 3x7= **21 sims**. ⟷ | RAD (Oper. Rads.) | ~ 6 h | ~20 |
| | | | | | GLOBE (1 km) | TERRA ML | | HR (High-res. Rads.) | ~6 h | ~700 |
| 2.8 km | CTRL-7 | 50 | Tiedtke Shallow | 1D TKE | | | | GPS | 10 min | Integr. |
| 500m | CTRL-2.8 | 80 | - | 3D TKE | | | | GPS-RAD | Combined instruments | |
| | | | | | | | | RAD-HR | | |
| | | | | | | | | GPS-RAD-HR | | |
| | | | | | | | | CTRL (No obs.) | | |

*Most description of results repeat a lot of information that can be read from the figures. This makes these description quite lengthy and somewhat difficult to read. For example, the beginning of section 4.2 is especially hard to follow. The paragraph ~445-450 also repeats a lot of information accessible in the table being discussed. It is suggested that the description of results be shortened or summarized wherever possible.*

We have shortened the description of the results wherever it was possible, aiming at providing clearer descriptions of the findings.

*Often figures found in the supplementary material will be referred to alongside the other figures. For example on line 377 we find "... western side of the Alps (Figs. 4b, S1b and S2b)." If the supplementary material will not be immediately available to the readers of the manuscript it is suggested that the supplementary figures not be referred to directly. If these figures are necessary to the comprehension of the text, they should be included in the manuscript.*

We have adapted the manuscript not to refer to the SM repeatedly. Only needed graphs are included in the manuscript that are sufficient to comprehend and validate the expressed results.

*Following are minor comments in the order that they appear in the manuscript*

If no explanation is given, the minor correction has been addressed in the revision.

*Line 133: Because of image compression, the red squares in figure 1b look a lot like circles.*

It is true, still thanks to the colour difference between operational soundings (blue triangles) and the high-resolution (red squares), we believe these two observation types are readily distinguishable by eye. Additionally, we have changed, in line 133, the word "squares" for "markers".

*Equation 3: Out of curiosity, what is the value of "s" being used? Does it change with the resolution of the model or the observations being assimilated? Should it?*

"s" is defined as correlation scale and provides a factor for attenuation of assimilation impact when spreading the information horizontally. "s" varies with altitude and is a parameter pre-defined in the model. For example, for humidity (q) and temperature (T) the correlation scale parameter in km is as follows for pressure levels between 1000 hPa and 50 hPa.

**Table 1. Correlation scales for temperature $s_T$ and humidity $s_q$ in [km] at the observation time as a function of pressure p in [hPa]. Obtained from the COSMO model documentation.**

| P (hPa) | 1000 | 850 | 700 | 500 | 400 | 200 | 150 | 100 | 50 |
|---|---|---|---|---|---|---|---|---|---|
| $s_T$, $s_q$ (km) | 58 | 66 | 75 | 83 | 83 | 91 | 100 | 100 | 100 |

For illustration, applying a $s_q = 83\ km$ for humidity at a 500 hPa implies that the weight of the observation for the horizontal spreading $w_{xy}$ is halved at a distance of 135 km from the observation's location.

The values in Tab. 1 are used for radiosondes but for GPS measurements a scaling of 45 % is applied, to account for the fact that GPS observations are typically much denser than radiosondes. The values of $s_T$ and $s_q$ are constant, regardless of the used model resolution.

We did not perform supplementary experiments varying this parameter as our main goal was to assess the added value of the observation types and their impact on model variables rather than assessing how model parameters could be fine-tuned. We believe such experiments would fall out the scope of the paper. Nevertheless, our interpretation is that adapting the values for $s_T$, $s_q$ is more sensible for different observation types than for model resolution, not to harm the information of neighbouring observations (as in the case of GPS). However, reasonable conflicts could arise from the use of a too large correlation scale for observations close to the surface in different resolutions. The better representation of the model's orography in a 2.8 km and a 500 m resolution could impose orographic boundaries that should be considered to truncate the too large horizontal spreading.

This aspect is discussed briefly in the revision in Sect. 2.2.1 The COSMO Model, the nudging scheme:

"*Horizontally, the spreading is performed using a second-order autoregressive function of the distance between the observation location and the target point (Δr) divided by correlation scale (s), see Eq. (3). The values of s range between 58 and 100 km, depending on the model level for radiosondes and are reduced by 45 % for GPS data to avoid conflicting neighbouring observations, given its larger surface coverage (Schraff and Hess, 2012), The correlation scale is invariant under resolution changes as in its operational set-up. The impact of adapting s, to different model resolutions is not investigated here, as this would be out of the scope of the paper. It is advised however, further testing of different values of the correlation scale for higher resolutions to address any potential conflicts of assimilated observations with e.g. an increased resolution of the surface model's orography.* "

*In figure 4d) we can clearly see artifacts caused by the inflow through the model boundaries. Visibly, it takes some time for the model's parametrizations to generate precipitation from the inflow through the boundaries. Presumably, some of the microphysical species being modeled are initialized at zero at the boundaries. While this does not seem to affect the main areas of interests for this study, this illustrates the difficulties associated with such high resolution forecasts. This phenomenon would probably be worth mentioning.*

We have included this observation in the manuscript, in the description of Fig.4.

"*Finally, combining all observation types for nudging (GPS-RAD-HR-500, Fig 4d) yields a structure similar of that of the RAD simulations but with a weaker precipitation increase (Fig. 4b). It is worth mentioning the existence of model artifacts in the eastern part of the domain (Fig. 4d, for instance), which evidence the difficulties of dynamically downscaled simulations in initializing the microphysical species at the boundaries. Even though this does not affect the conclusions of this study, it shows that some of the species are being initialized at zero.*"

*Being from North-America, all locations listed on this line except the Rhone valley were unknown to me. Perhaps adding letters or arrows could help readers from abroad to locate these places more easily?*

The location of these cities/regions is now shown in Fig.2.b.

*Line 334 : "no dynamic impacts": In the Canadian system, the assimilation of radar-inferred precipitation through latent heat nudging is shown (see paper references above) to reduce RMSE for upper-level winds by a few percent on average over a two-month verification period (~110 forecasts). One would not expect to be able to observe such a small signal on the model dynamics for only one precipitation event.*

This information has been included in the manuscript to provide further insights on how the nudging of thermodynamic profiles brough  a low impact on wind components for our case of study.

*line 363 - The blending of the different precipitation products certainly explains part of the smoothness of the satellite-based products. Large differences in sampling volumes should also be mentioned as a factor contributing to the observed differences.*

This remark has been included in Line 363

*Line 421: Altitude-based corrections can sometimes be significant, especially in mountainous terrain where the difference between the model terrain and observation height can be large. Do we know if this is the case here?*

We follow the procedure suggested by Bock and Parracho (2019), where stations with height differences (station altitude vs. altitude of selected grid point) larger than 500 m are dismissed from the calculations. The IWV corrections applied to the remainder stations ($dIWV/IWV = -4 \cdot 10^{-4} \cdot dh$) bring corrections that averaged in time and space are no larger than 0.2 %. For a specific date, after spatially averaging to all stations (within investigation domain RhoAlps) are of 1 % and for particular stations can be as large as ±20 %. These corrections are necessary, especially over complex terrain to consider  the height differences. However, for the results presented in Fig. 7 and Tab. 2 bring a marginal impact (~ 1 %), since the values presented are spatially and timely averaged.

The following table (Tab. R1) shows the values of these corrections for a specific time step in the simulation, for some of the GPS stations within the investigation domain.

**Tab. R1 IWV corrections applied, following Bock and Parracho (2019). The surface height of GPS stations so as the model surface level is shown, as well as the IWV values, and the relative and absolute differences.**

| Stat ID | Height GPS [m] | Height COSMO [m] | Height diff [m] | IWV GPS [mm] | IWV COSMO [mm] | IWV COSMO (corr) [mm] | Rel Var [%] | Abs Var [mm] |
|---|---|---|---|---|---|---|---|---|
| 3 | 279.5 | 274.5 | 5 | 29.3 | 30.9 | 30.9 | 0.0 | 0 |
| 305 | 268.5 | 287.2 | -18.7 | 21.9 | 24.1 | 24.3 | 0.8 | 0.2 |
| 363 | 20.7 | 11.5 | 9.2 | 33.2 | 32.1 | 31.9 | -0.6 | -0.2 |
| 364 | 494.2 | 515.8 | -21.6 | 32 | 31.9 | 32.2 | 0.9 | 0.3 |
| 372 | 593.2 | 1069.5 | -476.3 | 26.9 | 25.9 | 30.8 | 18.9 | 4.9 |
| 500 | 657.2 | 723.9 | -66.7 | 26.9 | 23.8 | 24.4 | 2.5 | 0.6 |
| 519 | 1827.7 | 2062.2 | -234.5 | 14.5 | 15.2 | 16.6 | 9.2 | 1.4 |
| 521 | 19.6 | 4.1 | 15.5 | 23.5 | 25 | 24.9 | -0.4 | -0.1 |
| 523 | 725.8 | 227 | 498.8 | 16.6 | 18.7 | 14.9 | -20.3 | -3.8 |
| 524 | 1019.3 | 691.4 | 327.9 | 18.3 | 20.8 | 18 | -13.5 | -2.8 |
| 525 | 1019.3 | 691.4 | 327.9 | 19.2 | 20.8 | 18 | -13.5 | -2.8 |
| 528 | 1369.8 | 1386.4 | -16.6 | 21.6 | 23.1 | 23.3 | 0.9 | 0.2 |
| 665 | 656.6 | 690.8 | -34.2 | 30.7 | 30.5 | 30.9 | 1.3 | 0.4 |
| 669 | 220.5 | 225.8 | -5.3 | 32 | 31.2 | 31.2 | 0.0 | 0 |
| 670 | 1025.7 | 1161.9 | -136.2 | 21.4 | 17.3 | 18.3 | 5.8 | 1 |
| 671 | 300.3 | 388.6 | -88.3 | 22.6 | 23.2 | 24 | 3.4 | 0.8 |
| 672 | 267.3 | 235.4 | 31.9 | 32.6 | 29.5 | 29.2 | -1.0 | -0.3 |
| 675 | 195 | 227.4 | -32.4 | 28.9 | 27.5 | 27.9 | 1.5 | 0.4 |
| 678 | 274.8 | 240.6 | 34.2 | 29.9 | 30.4 | 30 | -1.3 | -0.4 |
| 688 | 907.6 | 667.2 | 240.4 | 25.6 | 30.6 | 27.7 | -9.5 | -2.9 |
| 689 | 907.5 | 667.2 | 240.3 | 25.1 | 30.6 | 27.7 | -9.5 | -2.9 |

This information is briefly mentioned in the revision.

*"The correction is based on an empirical linear relationship between IWV biases and height differences (dh) following the equation $dIWV/IWV = -4 \cdot 10^{-4} \cdot dh$. Grid points with surface height differences larger than 500 m are dismissed. The average impact of these corrections does not exceed 1 % of IWV."*

*Figure 7: The black line for GPS is difficult to distinguish in this figure. Maybe use a thicker/dashed line style?*

We acknowledge that the GPS black line is hard to see, precisely because of the good performance of the runs with assimilated observations, that overlay the black line of the GPS. We have added a note "underneath the coloured lines" in Sect. 4.2 to make clear to the reader that the simulations with assimilated observations is underneath all the rest.

*Line 532: In other instances of the text, the great heterogeneity of the moisture field is mentioned as a source of complications. It seems reasonable to assume that this likely explains why high moisture content was measured by only one sounding.*

We have adapted the corresponding paragraph:

*"The reason why other soundings close to Nimes in time and space did not measure such a large humidity amount at 700 hPa is still unknown. The large spatial heterogeneity of this variable might have played a decisive role and its sampling has already been identified as a factor limiting heavy precipitation simulation (Khodayar et al., 2018). Unfortunately, no other humidity observations exist for that time and location (LIDAR, pressurized balloons or dropsondes). Another possible explanation is an ascent of the Nimes_0515 sounding through a precipitating system."*

*Section 4.3.1: The box plots shown in figure 10 show no obvious differences that would be statistically different between the various experiments. Since this section is quite detailed and the manuscript already long, it is suggested that this section be moved to the supplementary materials. If it is believed that the section should remain in the manuscript, lines ~560-575 should be reworked to improve readability.*

We have shortened the text to improve the readability. However, we believe that the explanation on the impact of the sounding on precipitation processes is relevant for the study. See "track_changes" for revision of the corrections.

**References**

Beck, H. E., van Dijk, A. I. J. M., Levizzani, V., Schellekens, J., Miralles, D. G., Martens, B. and de Roo, A.: MSWEP: 3-hourly 0.25global gridded precipitation (1979–2015) by merging gauge, satellite, and reanalysis data, Hydrology and Earth System Sciences, 21(1), 589–615, doi:10.5194/hess-21-589-2017, 2017.

Bock, O. and Parracho, A. C.: Consistency and representativeness of integrated water vapour from ground-based GPS observations and ERA-Interim reanalysis, Atmospheric Chemistry and Physics, 19(14), 9453–9468, doi:10.5194/acp-19-9453-2019, 2019.

Hodam, S., Sarkar, S., Marak, A. G. R., Bandyopadhyay, A. and Bhadra, A.: Spatial Interpolation of Reference Evapotranspiration in India: Comparison of IDW and Kriging Methods, Journal of The Institution of Engineers (India): Series A, 98(4), 511–524, doi:10.1007/s40030-017-0241-z, 2017.

Jacques, D., Michelson, D., Caron, J.-F. and Fillion, L.: Latent Heat Nudging in the Canadian Regional Deterministic Prediction System, Monthly Weather Review, 146(12), 3995–4014, doi:10.1175/mwr-d-18-0118.1, 2018.

Khodayar, S., Czajka, B., Caldas-Alvarez, A., Helgert, S., Flamant, C., Girolamo, P. D., Bock, O. and Chazette, P.: Multi-scale observations of atmospheric moisture variability in relation to heavy precipitating systems in the northwestern Mediterranean during HyMeX IOP12, Quarterly Journal of the Royal Meteorological Society, 144(717), 2761–2780, doi:10.1002/qj.3402, 2018.

Schraff, C. and Hess, R.: A Description of the Nonhydrostatic Regional COSMO-Model Part III: Data Assimilation, German Weather Service (DWD), P.O. Box 100465, 63004 Offenbach., 2012.

**Answers to Reviewer 2 - wcd-2021-2**

We would like to thank the anonymous reviewer for her/his valuable comments and corrections. We have accepted most of the remarks and included them in the revised version of the manuscript. In the following, we provide detailed answers to the questions/requests. In case our corrections need no further explaining, they have been directly included in the new version of the manuscript. The specific changes can be seen in the "track_changes" version.

**General comment**

*In this paper, the impact of assimilating GPS-ZTD data and sounding observations at low and high vertical resolution is evaluated on a case study of heavy precipitation. The COSMO model is employed at 3 different resolutions over 3 domains and observations are assimilated by a nudging technique. Verification of experiments is performed using several metrics, especially regarding precipitation. Regardless of the model resolution, only the assimilation of operational low vertical resolution radiosondes improves precipitation accuracy, while both high-resolution soundings and GPS observations have a negative impact. This is probably due to deficiencies in model physics and, for GPS, to lack of vertical information.*

*In my opinion, the topic is relevant and the experiments presented by the authors are interesting. The paper is well written and results are discussed in detail. However, I think that some aspects of the manuscript need to be improved, as reported in the following comments.*

**Major comments**

**High-resolution radiosondes thinning**

*High vertical resolution radiosondes (HR) are assimilated without performing any thinning or data reduction. As far as I understand, since HR vertical levels are much more than model levels (700 compared to 40-80), this means that HR observations are overweighted. I think that this point should be reported and discussed in the manuscript.*

We agree that this aspect should be discussed in the manuscript.

The nudging procedure of the COSMO model reads radiosonde reports as they are made available, and after quality and consistency checks, observations are either averaged over each model layer (temperature, wind) or vertically interpolated to the height of the mid model layer (humidity). This is done both for operational soundings (RAD) and high-resolution (HR). The improvement gained from the DA comes therefore from how the larger number of levels impacts the layer averages (wind, temperature) or vertical interpolations to the mid model layer (humidity). This aspect will be more relevant as the number of vertical levels is increased with finer model resolutions (40 levels in a 7 km set-up, 50 in 2.8 km and 80 for 500 m).

To bring this discussion in the manuscript the following changes are introduced in Sect. "Nudging of GPS and radiosondes" within section 2.2.1.

"*The COSMO nudging scheme only allows the assimilation of prognostic variables. In the case of the radiosondes, COSMO reads profiles of temperature, wind and humidity assigning all observations to a grid point in model space. Given that the grid points cannot correctly represent wavelengths of $2\Delta x$ or less, the assignment is performed with no interpolation in the horizontal direction (Schraff and Hess, 2012). The observations are averaged over each model layer for temperature and wind and interpolated to the representative height of each model level for humidity. Therefore, the higher the number of vertical model levels the more the assimilation will profit from higher vertical resolution in*

*the radiosondes*. *The impact of the analysis increments on the neighbouring grid points is controlled through lateral ($w_{xy}$), vertical ($w_z$) and temporal weights ($w_t$) through the equation $w_k = w_{xy} \cdot w_z \cdot w_t \cdot \varepsilon_k$, where $\varepsilon_k$ accounts for the quality and representativeness of the observation. At the exact time-space location of the observation $w_{xy}$, $w_z$ and $w_t$ are set to 1.*"

In sect 4.1 (discussion of Figure 5)

"*Nudging HR, similarly to GPS reduces the 24-hly precipitation amount resulting in worse scores for this metric. In this regard the higher vertical resolution of HR did not bring added value for this case study, compared to RAD.*"

In Sect. 4.3 (discussion of Figure 9)

"*This implies that in the RAD simulations the existing errors in COSMO regarding the underestimation of humidity at the LFT (see Fig. 8) and the need of excessive moisture to represent sufficient rain are compensated by this one sounding. This highlights the relevance and complications of targeted observations for DA. Moreover, it also highlights that, for this case study, the accurate location and timing of that one sounding were more relevant for precipitation simulation than the higher vertical resolution offered by the HR data set. Spatial distances of 60 km and temporal differences of 30 minutes are enough to miss/capture a crucial measurement of water vapour.*"

**Figure 5**
*Figure 5 is crucial to quantitatively assess the impact of the various experiments on precipitation accuracy. However, some aspects are not clear and should be discussed further. First of all, it should be explained how the 99th percentile of 3h precipitation is computed.*

We acknowledge that the manuscript needs further explanation on how the 99[th] percentile of the 3h precipitation aggregates are computed. To this end, we have extended Sect. 2.4 (Verification Metrics) with the following paragraph.

"*2.4.1 Percentile-99 of three hourly precipitation aggregates*

*We validate extreme precipitation intensity simulated by COSMO against MSWEP. To this end we upscale COSMO's grid to the MSWEP spatial resolution (0.1°) by means of bilinear interpolation. Then we obtain 3-hourly precipitation aggregates for the grid points within the investigation area. The 99-percentile is obtained from the sample of all 3-hourly precipitation intensities at each grid point during the day of precipitation i.e., for eight time steps during 24 September 2012.*"

This sub-section is adequately referenced within the description of Fig. 5 in section 4.1.

"*How these metrics are computed is introduced in Sect. 2.4.*"

*Moreover, how are the raingauges treated? For example, for the domain average, are they aggregated to the same grid of MSWEP to take into account spatial variability?*

We used the values of RG 24hly and 3hly precipitation aggregates at each station to obtain the spatial average (Fig.5a) and the 99-perc percentile (Fig.5b). Hence, in the old version of the manuscript, no interpolation to a common grid (0.1° from MSWEP) was carried out for RG. This, as pointed out, can pose problems regarding comparability between the different data sets.

To further extend this analysis, we have carried out interpolations of the RG 24hly and 3hly to the 0.1° MSWEP native grid by means of an Inverse Distance Weighting Method (Hodam et al., 2017). However, the interpolation has revealed spurious artifacts around the point stations and unrealistic

precipitation gradients with no agreement with the original RG distribution. See for example the unrealistic dotted features in Fig. R1a or the artificial precipitation gradients in the northwestern part of the domain in Fig. R1b

a)

[Figure]

b)

[Figure]

**Fig R1. Spatial distribution of 24hly precipitation on 24 September 2012 a) and 3hly precipitation on 24 September 2012 at 21:00 UTC. The represented data set are the RG after interpolation through Inverse Distance Weighting (IDW) to the MSWEP grid (0.1°).**

Therefore we have decided to restrict the verification to the MSWEP product to avoid the artifacts produced by the gridding of RG data. This decision is further supported by the good results of MSWEP in previous publications (Beck et al., 2017), the fact that it already contains station data and that it complies better with the spatial scale of the coarse-grained COSMO results.

*Finally note that the title of subplot "b" has to be swapped with that of subplot "c".*

Indeed, panels b) and c) within Fig. 5 were interchanged. This has been corrected in the new version of the manuscript.

**Minor comments**

*L56-58. In contrast to GPS, satellite and radar are claimed to not be all-weather observations. Regarding radar reflectivity, even if it is particularly useful in case of precipitation, it can be gainfully assimilated also in no-precipitating conditions to suppress spurious model rainfall (see for example Bick et al. (2016) and Gastaldo et al. (2021) for COSMO-LETKF, but the same holds for nudging schemes). About satellite observations, clear-sky observations have been assimilated for many years, but there are several studies dealing with the all-sky assimilation (see for example Geer et al. (2018) for a review). So, please explain more in detail what you mean.*

This statement was incorrect. Satellite and ground radars also measure atmospheric variables in cloud-precipitation situations. The intention was highlighting the advantages of GPS in measuring IWV as opposite to satellite products for the same variable. For example, IWV measurements from MODIS only provide IWV estimates in clear conditions, or cloudy areas, above the cloud tops. As opposite to GPS, that also in the presence of clouds can provide information of the IWV.

The statement in the introduction has been rephrased.

*"The advantages of this product are its high temporal resolution, that it is all-weather (provides IWV estimates in cloudy as well as clear sky situations), has large accuracy (Bock et al., 2016, Bock et al., 2019; Jones et al., 2019)"*

*L61-62. I am not sure that Davolio et al. (2017) restrict the correction to boundary layer. Looking at their Table 3, the moisture correction is smoothed in the boundary layer.*

Yes, this is correct. In the Davolio et al., (2017) paper it is explained: "The main role of the parameter $\nu(k)$ is to limit the specific humidity adjustment in the boundary layer, in order to avoid too unstable profiles that can produce excessive convective activity".

Indeed, the correction is not restricted in the boundary layer, and it is truncated only at a height of 8 km.

The information has been corrected in the manuscript.

*L190-231 Several symbols employed in the equations and in the text are not explicitly defined like, for instance, F, x, t, xk in eq 1, all variables in eq 2, ps and Tm at line 220. It is true that most symbols are easy to interpret, but I think it would be more clear to define all of them.*

The variables have been defined for each equation in the new version of the manuscript (see the "track_changes" supplement).

*L275-280. Some aspect are not clear to me. Are you computing FSS employing moving boxes consisting of 18 grid points? Why 18 is the maximum number of grid points in the RhoAlps domain? Please rephrase these lines.*

There was a mistake in this explanation. We compute FSS using moving boxes of neighbour length (N=20), not 18. This means that the fractions of precipitation ($f = n_{precip}/n_{tot}$.) for the model ($f_{mod}$) and the observations ($f_{obs}$) are computed using 2*20+1 grid points in both directions (a total of $n_{tot} = 1681$ grid points). This choice of neighbour length N is selected given the fact that the largest skill of the forecast is given when N is the largest possible. Provided the shortest dimension of the investigation area RhoAlps is $n_{lat} = 42$, N=20 is the maximum neighbour length possible, to comply with $n = 2N + 1$. This is what is defined in Roberts and Lean (2008) as Asymptotic Fractions Skill Score (AFSS), that theoretically would have value of 1 in the case of no bias between the model and the

observations. This is the upper limit of the forecast skill. On the other hand the lowest limit is defined by the target FSS defined as $FSS_{target} = 0.5 + f_{obs}/2$ .

This explanation has been reworked to provide a better explanation on how the FSS is computed. It now shows as follows:

*"2.4.3 Fractions Skill Score (FSS)*

*The FSS provides an estimate of the agreement in the fraction of surface affected by precipitation between observations and simulations. After coarse-graining the simulations to the resolution of the observations (MSWEP, 0.1°), each grid point within the investigation area (both for observations and simulations) is given a value of 1 if precipitation is larger than 20 mmd-1 and 0 to the remainder grid points. We selected this precipitation threshold to be able to have defined precipitation structures within the investigation area (Roberts and Lean, 2008; Skok et al., 2016). We obtain the fractions of area, affected by precipitation in the model ($f_{mod}$) and the observations ($f_{obs}$) for moving sub-boxes. The fractions are computed as the ratio of the number of grid points with value 1 ($n_{precip}$) divided by the total number of grid points ($n_{tot}$), of the moving sub-boxes ($f = n_{precip}/n_{tot}$). The size of the sub-boxes is defined by the Neighbour Length (N). We choose the maximum possible N to guarantee the largest skill of the forecast. The maximum N is defined by the number of grid points in the shortest dimension of the investigation area. In our case this is the latitude dimension ($n_{lat} = 42$). N has to fulfil the condition $n_{lat} = 2N − 1$, hence the neighbour length (N) of the moving boxes is 20. The FSS is computed as shown in Eq. 6.*

$$FSS = 1 - \frac{\frac{1}{M}\sum_{i=1}^{M}(f_{mod}-f_{obs})^2}{\frac{1}{M}(\sum_{i=1}^{M}f_{mod}^2+\sum_{i=1}^{2}f_{obs}^2)} \qquad [6]$$

*Where M is the number of sub-boxes. Eq. 6 corresponds to what is defined in Roberts and Lean (2008) as Asymptotic Fractions Skill Score (AFSS). This asymptotic value is reached when the number of neighbours is the largest. It provides the largest skill of the verification and if there is no bias between the model and the observations AFSS equals one. On the other hand the lower limit of the model's skill is defined by the target FSS defined as $FSS_{target} = 0.5 + f_{obs}/2$ and is denoted by a dashed line in Fig.5c. Below this threshold the forecast has no skill."*

*L294 Some cities are reported here. They should be indicated on the map or, at least, geographical coordinates have to be specified.*

The cities are shown now in Fig. 2b. and Fig.1b

*Figure 2, 3 and 4. When a nonlinear colorbar is adopted, as for precipitation here, all bin extremes should be specified.*

This has been corrected in the new version of the manuscript.

*L311-312. MSWEP clearly underestimates precipitation over Liguria region compared to RG, This should be reported. Moreover, this may also be taken into account for the subsequent qualitative verification (Fig. 3 and 4).*

The underestimation over Liguria has been noted in the new version of the manuscript in Sect. 3:

*"Overall, MSWEP represents overall well the event over the RhoAlps area albeit clear differences in structure due to the coarser resolution of ~ 10 km, an underestimation over the Liguria area and an overestimation north of the Rhone valley and over the Alps , compared to RG."*

*L464-465. As in L294, some cities are reported here. They should be indicated on the map or, at least, geographical coordinates have to be specified.*

The locations of the cities are now specified in Fig.2.b and 1.b.

**References**

Beck, H. E., van Dijk, A. I. J. M., Levizzani, V., Schellekens, J., Miralles, D. G., Martens, B. and de Roo, A.: MSWEP: 3-hourly 0.25global gridded precipitation (1979–2015) by merging gauge, satellite, and reanalysis data, Hydrology and Earth System Sciences, 21(1), 589–615, doi:10.5194/hess-21-589-2017, 2017.

Hodam, S., Sarkar, S., Marak, A. G. R., Bandyopadhyay, A. and Bhadra, A.: Spatial Interpolation of Reference Evapotranspiration in India: Comparison of IDW and Kriging Methods, Journal of The Institution of Engineers (India): Series A, 98(4), 511–524, doi:10.1007/s40030-017-0241-z, 2017.

Roberts, N. M. and Lean, H. W.: Scale-Selective Verification of Rainfall Accumulations from High-Resolution Forecasts of Convective Events, Monthly Weather Review, 136(1), 78–97, doi:10.1175/2007mwr2123.1, 2008.

Skok, G. and Roberts, N.: Analysis of Fractions Skill Score properties for random precipitation fields and ECMWF forecasts, Quarterly Journal of the Royal Meteorological Society, 142(700), 2599–2610, doi:10.1002/qj.2849, 2016.